# Cosmogenic nuclide and solute flux data from central Cuban rivers emphasize the importance of both physical and chemical mass loss from tropical landscapes

**Mae Kate Campbell**[1,2], **Paul R. Bierman**[2,3], **Amanda H. Schmidt**[4], **Rita Sibello Hernández**[5], **Alejandro García-Moya**[5], **Lee B. Corbett**[3], **Alan J. Hidy**[6], **Héctor Cartas Águila**[5], **Aniel Guillén Arruebarrena**[5], **Greg Balco**[7], **David Dethier**[8], and **Marc Caffee**[9,10]

[1]Department of Geology, University of Vermont, Burlington, VT 05405, USA

[2]Gund Institute for Environment, University of Vermont, Burlington, VT 05405, USA

[3]Rubenstein School of the Environment and Natural Resources, the University of Vermont, Burlington, VT 05405, USA

[4]Department of Geosciences, Oberlin College, Oberlin, OH 44074, USA

[5]Centro de Estudios Ambientales de Cienfuegos, Departamento de Estudio de la Contaminación Ambiental. AP 5, 59350, Ciudad Nuclear, Cienfuegos, Cuba

[6]Atmospheric, Earth, and Energy Division, Lawrence Livermore National Laboratory, Livermore, CA 94550, USA

[7]Berkeley Geochronology Center, Berkeley, CA 94709, USA

[8]Department of Geosciences, Williams College, Williamstown, MA 01267, USA

[9]Department of Physics and Astronomy, Purdue University, West Lafayette, IN 47907, USA

[10]Department of Earth, Atmospheric, and Planetary Sciences, Purdue University, West Lafayette, IN 47907, USA

**Correspondence:** Amanda H. Schmidt (aschmidt@oberlin.edu)

**Abstract.** We use 25 new measurements of in situ produced cosmogenic $^{26}$Al and $^{10}$Be in river sand, paired with estimates of dissolved load flux in river water, to characterize the processes and pace of landscape change in central Cuba. Long-term erosion rates inferred from $^{10}$Be concentrations in quartz extracted from central Cuban river sand range from 3.4–189 Mg km$^{-2}$ yr$^{-1}$ (mean 59, median 45). Dissolved loads (10–176 Mg km$^{-2}$ yr$^{-1}$; mean 92, median 97), calculated from stream solute concentrations and modeled runoff, exceed measured cosmogenic-$^{10}$Be-derived erosion rates in 18 of 23 basins. This disparity mandates that in this environment landscape-scale mass loss is not fully represented by the cosmogenic nuclide measurements.

The $^{26}$Al / $^{10}$Be ratios are lower than expected for steady-state exposure or erosion in 16 of 24 samples. Depressed $^{26}$Al / $^{10}$Be ratios occur in many of the basins that have the greatest disparity between dissolved loads (high) and erosion rates inferred from cosmogenic nuclide concentrations (low). Depressed $^{26}$Al / $^{10}$Be ratios are consistent with the presence of a deep, mixed, regolith layer providing extended storage times on slopes and/or burial and extended storage during fluvial transport. River water chemical analyses indicate that many basins with lower $^{26}$Al / $^{10}$Be ratios and high $^{10}$Be concentrations are underlain at least in part by evaporitic rocks that rapidly dissolve.

Our data show that when assessing mass loss in humid tropical landscapes, accounting for the contribution of rock dissolution at depth is particularly important. In such warm, wet climates, mineral dissolution can occur many meters below the surface, beyond the penetration depth of most cosmic rays and thus the production of most cosmogenic nuclides. Our data suggest the importance of estimating solute fluxes and measuring paired cosmogenic nuclides to better understand the processes and rates of mass transfer at a basin scale.

# 1 Introduction

Cosmogenic nuclide concentrations of river sand have been used to quantify rates of landscape change (often termed erosion rates) since the 1990s (Brown et al., 1995; Granger et al., 1996; Bierman and Steig, 1996; Portenga and Bierman, 2011; Codilean et al., 2018). Accurately establishing long-term rates of change provides an important context for understanding the effects of human activity on erosion (Reusser et al., 2015; Nearing et al., 2017) and for other common applications of cosmogenic nuclides at the basin scale, such as quantifying the effect of tectonics (Scherler et al., 2014), climate (Marshall et al., 2017), and base-level change (Reinhardt et al., 2007) on rates of landscape change over time.

$^{10}$Be-derived rates of landscape change at a drainage basin scale are often implicitly assumed to reflect both physical and chemical mass loss, the sum of which is termed denudation (Regard et al., 2016). However, this assumption is only valid if all mass loss from the landscape occurs within the uppermost 1–2 m of Earth's surface, the penetration depth of the cosmic-ray neutrons responsible for producing most cosmogenic nuclides via spallation reactions (Bierman and Steig, 1996). Deeper mass loss by rock dissolution remains largely undetected by cosmogenic nuclide analysis. Failure to account for rock dissolution at depth and the export of mass as dissolved load below the spallation-dominated nuclide production zone ($\sim 2$ m) may bias cosmogenic-nuclide-derived estimates of denudation (Small et al., 1999; Riebe et al., 2001a; Dixon et al., 2009a) on the low side. Incorrectly determined erosion rates can derail attempts to understand landscape evolution, soil production, and climate interaction with surface processes (Riebe et al., 2003).

Rock dissolution at depth is an important process in areas with significant groundwater–rock interactions; connecting denudation rates to landscape change requires consideration of this process. This includes any landscape where the physical removal of mass is slow, allowing for prolonged water–rock interactions, such as low-relief landscapes (Ollier, 1988). Some landscape characteristics facilitate or are the result of extensive water–rock interaction: thick saprolite (Dixon et al., 2009a), extensively jointed and/or fractured bedrock (Ollier, 1988), and readily soluble rocks, including carbonate (Pope, 2013) and evaporite deposits. Conditions in the humid tropics favor prolonged and extensive water–rock interaction and include the absence of recent glaciation (Modenesi-Gauttieri et al., 2011), the presence of active groundwater flow systems year-round (Ollier, 1988), and large amounts of precipitation.

Rock dissolution rates in the tropics can be among the highest globally (Pope, 2013); yet, global compilations of cosmogenic nuclide data from river sand suggest that rates of landscape change in the tropics are slower than in most other climate zones (Portenga and Bierman, 2011). This dichotomy is consistent with cosmogenic rates significantly

underestimating landscape denudation in areas where deep rock dissolution is ubiquitous.

Only a few studies focused in the tropics compare nuclide-derived rates to measurements of dissolved load flux in streams (e.g., Salgado et al., 2006; Hinderer et al., 2013; Regard et al., 2016). As the use of cosmogenic nuclides to measure rates of landscape change in the tropics expands (e.g., Cherem et al., 2012; Barreto et al., 2013; Derrieux et al., 2014; Mandal et al., 2015; Sosa Gonzalez et al., 2016a; Jonell et al., 2017), considering the potential influence of rock dissolution at depths below the production of most cosmogenic nuclides becomes more important.

Here, we present measurements of in situ $^{26}$Al and $^{10}$Be in riverine quartz, along with estimates of dissolved loads, in humid, tropical central Cuba (Bierman et al., 2020). With these data, we explore the relationships between cosmogenic nuclide concentrations, dissolved load fluxes, and landscape-scale parameters at a basin scale in a humid tropical location where mass is being lost from the landscape by multiple different processes from a variety of rock types. We characterize the rates and processes by which the Cuban landscape is changing and place these data in a global context. Our findings illustrate the importance of considering rock dissolution when using cosmogenic nuclides to assess rates of landscape change in areas with the potential for significant mass loss by solution at depth and provide a geologic baseline for assessing the impact of human actions on the Cuban landscape.

# 2 Background

Terminology referring to mass loss from watersheds has been applied ambiguously in the past and can be confusing. Here, we refer to the tempo of landscape mass loss calculated from $^{26}$Al and $^{10}$Be concentrations as erosion rates; these rates include all processes (physical and chemical) removing mass within $\sim 2$ m of Earth's surface. We refer to rates of landscape mass loss inferred from measurements of stream water chemistry, convolved with estimates of annual runoff volumes, as rock dissolution rates. We use the term denudation to refer to total mass loss from sampled catchments. All of these rates are expressed in terms of mass per time per area ($\mathrm{Mg\,km^{-2}\,yr^{-1}}$), which can be converted to depth over time by assuming a rock density.

## 2.1 Quantifying basin mass loss with cosmogenic nuclides: approaches and limitations

Landscape-scale denudation occurs through both physical removal of mass (erosion) and chemical dissolution of minerals in rocks. Sediment produced from eroding bedrock travels downslope towards base level, whereas rock dissolution moves mass in solution from the landscape to rivers and then to the ocean. Measurement of cosmogenic nuclides in river sediment can be used to infer the spatially averaged erosion rate of a drainage basin (Brown et al., 1995; Granger et al.,

1996; Bierman and Steig, 1996). In a basin that is steadily eroding, the concentration of cosmogenic nuclides in a sediment sample reflects the rate at which overlying mass at and near the surface was removed as the material was exhumed through both physical mass loss and rock dissolution (Lal, 1991). Cosmogenic erosion rates are equivalent to denudation rates if, and only if, rock dissolution only occurs within 1–2 m of the surface – the depth of penetration for neutrons which produce most cosmogenic nuclides. If rock dissolution occurs below the neutron penetration depth, erosion rates calculated from measured nuclide concentrations will underestimate denudation.

Measuring multiple cosmogenic nuclides with different half-lives in the same sample can provide more information on the near-surface history of surface materials, such as soil mixing depth and residence time (Lal and Chen, 2005), as well as sediment storage within the watershed (Granger and Muzikar, 2001). The production ratio of $^{26}$Al / $^{10}$Be at Earth's surface at middle and low latitudes is constrained by measurements and nuclear physics (Nishiizumi et al., 1989; Balco et al., 2008). If sediment that has accumulated cosmogenic nuclides is buried such that production is diminished over $> 10^5$ yr, the production ratio decreases because $^{26}$Al decays more rapidly than $^{10}$Be. Vertical soil mixing intermittently buries sediment grains, suppressing the $^{26}$Al / $^{10}$Be ratio in sediment shed from the landscape surface (Makhubela et al., 2019).

Paired cosmogenic isotope concentrations are visualized using a two-isotope diagram; the $y$ axis is the $^{26}$Al / $^{10}$Be ratio and the $x$ axis is the concentration of $^{10}$Be with normalization based on the production rate of nuclides at the sample site (Klein et al., 1986; Granger, 2006). Sediment samples that have experienced constant exposure with no erosion, or constant exposure under steady-state erosion, will plot within an enclosed region along the top of the diagram; samples that have experienced more complex exposure histories, including burial during or after cosmic-ray exposure, will plot below this region. Such complex histories could include development of a vertically mixed surface layer (Bierman et al., 1999; Lal and Chen, 2005) as well as extended burial during transport down slopes and in and along rivers.

Using measurements of cosmogenic nuclides to determine basin-averaged denudation rates requires the assumptions that mass loss from the basin is in steady state, that the mineral used for cosmogenic nuclide measurements is uniformly distributed throughout the watershed, and that denudation occurs within the penetration depth of most cosmic rays, which is the upper several meters of Earth's surface (Bierman and Steig, 1996). The grain size fraction selected for cosmogenic nuclide analysis must also be representative of the grain size distribution of sediment being produced on slopes (Lukens et al., 2016), although in many landscapes cosmogenic nuclide concentrations do not vary by sediment grain size.

Erosion rates calculated from cosmogenic nuclides can be inaccurate if these assumptions are violated. Rock dissolu-

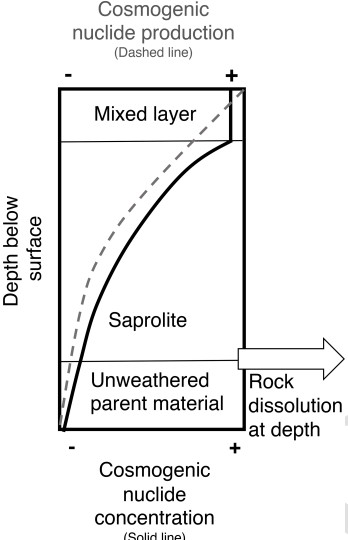

**Figure 1.** Conceptual diagram showing cosmogenic nuclide production and concentration in a column of soil, saprolite, and rock. The dashed line shows decreasing production of cosmogenic nuclides with depth; the solid line shows nuclide concentration with depth, and the white arrow represents mass loss by solution below the depth of significant cosmogenic nuclide production.

tion can leave sediment enriched in resistant mineral phases, such as zircon, titanite, and quartz – the mineral in which $^{26}$Al and $^{10}$Be are most commonly measured (Riebe and Granger, 2013). Such enrichment produces underestimates of long-term denudation rates unless accounted for because the enriched mineral will have a longer residence time relative to the surrounding regolith (Riebe et al., 2001a; Ferrier and Kirchner, 2008). Calculations of denudation rates from cosmogenic nuclide concentrations also rely on the assumption that mass loss is occurring primarily through surface lowering; however, some rock dissolution and thus some transfer of mass from rock to groundwater solutions occur below the depth of most cosmogenic nuclide production (Fig. 1; Small et al., 1999; Dixon et al., 2009a; Riebe and Granger, 2013). In areas with significant rock dissolution at depth, denudation rates inferred from cosmogenic nuclides underestimate denudation because some mass loss occurs below the depth of most nuclide production.

## 2.2 Chemical weathering corrections to cosmogenically determined mass loss rates

Although the importance of accounting for loss of mass by chemical weathering (rock dissolution) when calculating cosmogenic erosion rates has been recognized (Small et al., 1999; Riebe et al., 2001a; Dixon et al., 2009a; Riebe and Granger, 2013), few studies incorporate rock dissolution information or apply correction factors to cosmogenic-nuclide-derived rates. In the tropics, some studies compare export

rates from dissolved loads in streams to cosmogenically derived erosion rates, but those studies have considered these two metrics of landscape change separately (von Blanckenburg et al., 2004; Salgado et al., 2006; Hinderer et al., 2013). Other studies use the measurement of insoluble elements in bedrock, saprolite, and soil to quantify quartz enrichment through the weathering process and calculate correction factors that account for the influence of rock dissolution at and near the surface (Small et al., 1999; Riebe et al., 2001a), at depth (Dixon et al., 2009b), or both (Riebe and Granger, 2013).

Of studies that do correct for the influence of chemical weathering when calculating cosmogenic-nuclide-derived rates of erosion, the Riebe and Granger (2013) chemical erosion factor (CEF) method or earlier quartz enrichment factor method (Riebe et al., 2001a) is often used (Regard et al., 2016). Calculating a CEF requires measurements of soil thickness and density, as well as determining the concentration of the mineral used in cosmogenic nuclide measurements (commonly quartz) and an insoluble element (commonly Zr) in numerous samples of soil, saprolite, and unweathered bedrock. The method is underpinned by the assumption that chemical mass loss is occurring exclusively in well-mixed regolith and deep saprolite (Riebe and Granger, 2013). Erosion rates calculated from cosmogenic nuclide measurements can be multiplied by the CEF to correct for the effects of chemical mass loss (Riebe and Granger, 2013). Chemical erosion factors reported in tropical environments include a CEF of 1.79 in Puerto Rico (Riebe and Granger, 2013) and 3.2 in Cameroon (Regard et al., 2016), demonstrating how significantly cosmogenic-nuclide-derived estimates of erosion can underestimate total denudation rates by not accounting for the effects of deep rock dissolution.

## 3  Study area

Cuba is the largest Caribbean island ($\sim 110\,000\,\mathrm{km^2}$) and is situated along the boundary between the Caribbean and North American plates. Reflecting this complex tectonic setting, Cuban geology is varied and includes silicate, carbonate, and evaporite rocks (Pardo, 2009). Lithologies include marine deposits, accreted volcanic terrains, passive-margin sediments, and obducted ophiolite, all unconformably overlain by slightly deformed autochthonous coarse clastic sediment and limestone (Iturralde-Vinent et al., 2016).

The Cuban landscape features a mountainous central spine (600–1970 m) descending into low-relief coastal plains, except along portions of the southern coast where mountains meet the sea. This drainage divide parallels Cuba's east–west orientation, creating rivers that travel relatively short distances from headwaters to base level (Galford et al., 2018). Cuba's climate is tropical wet and dry, with a mean annual temperature of 24.5 °C and average annual precipitation of 1335 mm yr$^{-1}$. The climate is highly seasonal; $\sim 80\,\%$ of this precipitation is delivered during the wet season from May–October (Llacer, 2012).

Centuries of agriculture have heavily altered the Cuban landscape (Whitbeck, 1922). Prior knowledge of mass loss at the basin scale is limited to measurements of suspended sediment discharge for short periods between 1964 and 1983 for 32 Cuban rivers (Pérez Zorrilla and Ya Karasik, 1989) and measurements of dissolved loads in five limestone basins with karst (Pulina and Fagundo, 1992). In central Cuba, underlying basin rock type is the primary control on surface water geochemistry (Betancourt et al., 2012), a finding supported by geochemical analyses of river waters from the same basins sampled in this study (Bierman et al., 2020). Dissolved load fluxes carried by Cuban rivers (Bierman et al., 2020), and rock dissolution rates inferred from these fluxes, are consistent with rates reported for other Caribbean islands (Dominica, Guadeloupe, and Martinique from Rad et al., 2013, and Puerto Rico from White and Blum, 1995) and high compared to global data compiled by Larsen et al. (2014a).

## 4  Methods

### 4.1  Field methods

We collected detrital sediment ($n = 26$) from the beds of active river channels in central Cuba, representing a variety of basin sizes, average slopes, and lithologies (Fig. 2; Table S1 in the Supplement). Channel morphologies varied, but most streams were incised, and many had exposed bedrock (see Bierman et al., 2020, for photos and descriptions of select field sites). At each site we collected samples for water chemistry analysis and measured channel parameters, including width, depth, and discharge at time of sampling.

### 4.2  Lab methods

We prepared samples for cosmogenic analysis and extracted beryllium and aluminum following the methodology of Corbett et al. (2016). We sieved bulk sediment samples in the lab and used the 250–850 µm grain size fraction for all samples, except for CU-120, which also includes finer material (63–250 µm) due to low quartz content. Sediment samples were chemically etched to purify quartz and remove meteoric $^{10}$Be (Kohl and Nishiizumi, 1992). A total of 24 samples yielded sufficient quartz for analysis. We measured quartz yields for all but one sample (CU-120) by recording the mass of sediment before and after dilute acid etching (Fig. S1).

We extracted $^{26}$Al and $^{10}$Be at the National Science Foundation/University of Vermont Community Cosmogenic Facility using $\sim 5$–43 g of quartz per sample (mean 24 g). We added $\sim 250$ µg of $^{9}$Be to each sample using two different in-house-made carriers (Table S5); the first batch used a low-ratio carrier made from beryl; subsequent batches used a dilution of low-ratio commercial SPEX carrier. We added Al to samples with insufficient total Al using a commercial SPEX

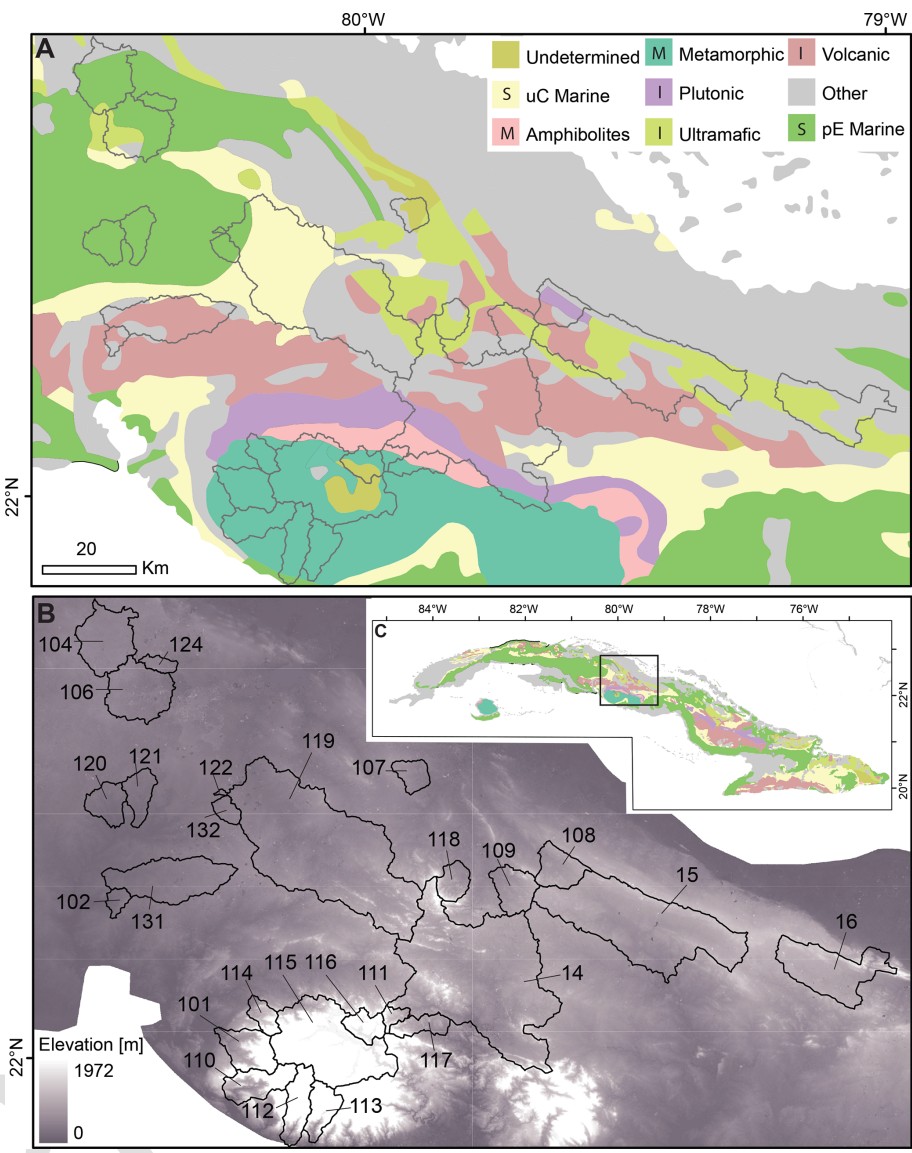

**Figure 2.** Maps showing underlying the basin geology (**a**; French and Schenk, 2004), elevations (**b**; LP DAAC, 2022), and location of the study area within the island of Cuba (**c**). The legend for panel (**a**) includes the category of the rock units in terms of sedimentary (S), igneous (I), and metamorphic (M). Note that the two marine units in panel (**a**) are separated because they have different river water chemistry.

ICP standard in order to reach a total Al mass of $\sim 1500\,\mu g$ (Table S6). Samples were processed in batches of 12, each of which included at least one blank, and two batches included one quality control standard each (Corbett et al., 2019).

$^{10}$Be / $^9$Be and $^{26}$Al / $^{27}$Al TS1 measurements ($n = 26$, including two duplicates) were made by accelerator mass spectrometry (AMS) at the Purdue Rare Isotope Measurement Laboratory (PRIME). $^{10}$Be ratios were normalized against standard 07KNSTD3110 with an assumed ratio of $2850 \times 10^{-15}$ (Nishiizumi et al., 2007), and $^{26}$Al / $^{27}$Al measurements were normalized against standard KNSTD with an assumed ratio of $1818 \times 10^{-15}$ (Nishiizumi, 2004). Full laboratory replicate sample preparations and measurements

of $^{26}$Al and $^{10}$Be agree to within $< 2\,\%$, with the exception of the $^{10}$Be replicate of CU-016, which was leached between the two lab analyses, and thus we use the replicate data only for that sample (Table S7; $n = 2$). We corrected Be measurements by carrier type, since samples were prepared using different carriers; we use the average of two process blanks ($1.91 \pm 1.01 \times 10^{-15}$; 1 SD) to correct 10 samples and the average of four process blanks ($4.02 \pm 1.00 \times 10^{-15}$; 1 SD) for the remaining samples (Table S3). We corrected Al measurements using the average of four process blanks ($4.97 \pm 2.94 \times 10^{-15}$; Table S4). We subtracted blank ratios from sample ratios and propagated uncertainties in quadrature.

## 4.3   Analytical methods

We extracted drainage basins and then calculated basin slopes and effective elevations (Portenga and Bierman, 2011) using the ASTER Global Digital Elevation Model (LP DAAC, 2022), determined underlying basin rock types from the USGS Caribbean layer (French and Schenk, 2004), and utilized precipitation data from the WorldClim dataset (Hijmans et al., 2005) to estimate basin-specific mean annual precipitation (MAP).

We calculated erosion rates using version 3 of the online erosion rate calculator originally described by Balco et al. (2008) and subsequently updated (wrapper: 3.0, erates: 3.0, muons: 3.1, validate: validate_v2_input.m – 3.0 consts: 2020-08-26) using the effective elevation (Portenga and Bierman, 2011) calculated for the basin upstream of the sample collection point, a sample thickness of 0 cm, a rock density of $2.6 \, \mathrm{g \, cm^{-3}}$, and assuming no topographic shielding across this low-relief landscape. We report erosion rates using the Stone–Lal production scaling scheme.

For samples with the highest $^{10}$Be concentrations ($n = 4$), we also measured the concentration of cosmogenic $^{21}$Ne in quartz to further characterize exposure history (Table S10). Neon isotope measurements were made at the Berkeley Geochronology Center on aliquots of the same purified quartz samples used for $^{26}$Al / $^{10}$Be analysis. They were done by vacuum degassing and noble gas mass spectrometry using the method described in Balter-Kennedy et al. (2020) and Balco and Shuster (2009).

We used measurements of dissolved loads in stream water (Bierman et al., 2020) and modeled annual flows from GLOH2O (Beck et al., 2015, 2017) to calculate rock dissolution rates for the 25 basins where we were able to collect water samples. To account for the wide range of lithologies in our upstream watersheds, including some with evaporites, we modified the approach used by Erlanger et al. (2021) (Fig. S2). We removed ions deposited as atmospheric inputs based on published data on dissolved concentrations in Cuban rainfall (Préndez et al., 2014). We then determined evaporite weathering rates by balancing Na with Cl and Ca with $SO_4$. The remaining Na was used to determine the silicate contribution of Mg and Ca by using an assumed ratio of Na / Mg of 0.25 and Na / Ca of 0.35 (Erlanger et al., 2021). Silicate weathering rates were calculated as the total of $SiO_2$ and $HPO_4$ assumed to result from silicate weathering. Finally, we balanced the remaining Mg and Ca with bicarbonate to determine carbonate weathering rates.

Considering a variety of landscape-scale metrics, we explored the relationship between $^{10}$Be-derived erosion rates and calculated rock dissolution (total and silicate, carbonate, and evaporite) rates using linear correlations and their associated $p$ values. All reported means of sample populations are arithmetic.

## 5   Results

Quartz sand, isolated from central Cuban river sediment, has high concentrations of cosmogenic nuclides (0.41 to $12.6 \times 10^5$ atoms g$^{-1}$ $^{10}$Be and 0.27 to $5.9 \times 10^6$ atoms g$^{-1}$ $^{26}$Al). $^{26}$Al / $^{10}$Be ratios (Fig. 4, Table 1) vary considerably, ranging from 3.65–8.36 (mean $5.72 \pm 1.14$, median 5.83). A total of 16 of 24 samples plot below the window defined by continuous exposure and steady erosion on the two-isotope diagram (Fig. 5). Because these $^{26}$Al / $^{10}$Be data indicate significant burial of quartz during and/or after exposure, many central Cuban drainage basins do not meet the assumption of insignificant nuclide decay inherent in calculations of erosion rates from cosmogenic nuclide concentrations in detrital sediment (Bierman and Steig, 1996). To minimize the impact of violating this assumption, we compare erosion rates based only on the longer-lived nuclide, $^{10}$Be, with landscape-scale metrics and dissolved loads. The $^{10}$Be rates, because they cannot properly account for loss of nuclides during burial for samples with depressed $^{26}$Al / $^{10}$Be ratios, are overestimates of the true rate of erosion but not necessarily denudation.

Erosion rates (Table S8), calculated from measured concentrations of $^{10}$Be (Table S7), differed considerably between sites. $^{10}$Be-derived erosion rates (Fig. 3) range from 3.4–189 Mg km$^{-2}$ yr$^{-1}$ (mean $59 \pm 52$, median 45). Considered to be bedrock lowering rates by assuming a bedrock density of $2.6 \, \mathrm{g \, cm^{-3}}$, these are 1.3–73 m Myr$^{-1}$ (mean $= 23 \pm 20$, median $= 17$). $^{10}$Be-derived erosion rates in central Cuba are weakly and positively correlated with mean annual precipitation and slope (Fig. 6). Quartz yields for the samples we analyzed varied widely (0.5 %–60 %, mean 20 %, median 17 %) but were not significantly correlated ($p \leq 0.05$) with any basin-scale variables or analytic results (Table S1, Fig. S1).

Rock dissolution rates (Fig. 3) range from 10–176 Mg km$^{-2}$ yr$^{-1}$ (mean $92 \pm 39$, median 97) and are higher than $^{10}$Be-derived erosion rates in 18 of the 23 basins in which we were able to make both measurements. The median rock dissolution rate is 2.15 times higher than the median $^{10}$Be-derived erosion rate. Rock dissolution rates and $^{10}$Be-derived erosion rates are not correlated (Fig. 6). However, when total rock dissolution rates are partitioned into silicate, evaporite, and carbonate rates, then the silicate dissolution rate is weakly positively correlated with $^{10}$Be-determined rates of erosion ($r^2 = 0.18$, $p < 0.05$, Table 2). Rock dissolution rates are not separable by dominant basin lithology. $^{10}$Be-inferred erosion rates are metamorphic > sedimentary and igneous > sedimentary (Fig. 4).

There is lithological dependence of $^{10}$Be-derived erosion rates and the ratio of rock dissolution to $^{10}$Be-derived erosion rates at the basin scale (Fig. 5). $^{10}$Be-derived erosion rates of sedimentary rocks were lower than other rock types ($p = 0.01$). Samples with the lowest $^{26}$Al / $^{10}$Be ratios and

**Table 1.** Summary of central Cuban drainage basin data.

| Sample | Lithology | Latitude (°N) | Longitude (°W) | Slope (°) | Area (km²) | MAP (mm yr⁻¹) | $^{10}$Be erosion rate (Mg km⁻² yr⁻¹) | ± | $^{26}$Al erosion rate (Mg km⁻² yr⁻¹) | ± | $^{26}$Al / $^{10}$Be | ± | Total diss (Mg km⁻² yr⁻¹) | diss / $^{10}$Be erosion | Max rate ($^{10}$Be + diss) |
|---|---|---|---|---|---|---|---|---|---|---|---|---|---|---|---|
| CU-014 | Igneous | 22.0662 | −79.7962 | 3.2 | 730 | 1456 | 163.0 | 14.6 | 161.0 | 21.2 | 7.15 | 0.64 | 71.2 | 0.44 | 234.2 |
| CU-015 | Igneous | 22.1485 | −79.4231 | 1.5 | 458 | 1362 | 64.5 | 5.7 | 71.4 | 8.3 | 6.32 | 0.38 | 137.7 | 2.13 | 202.2 |
| CU-016 | Igneous | 22.2090 | −79.0172 | 3.0 | 177 | 1491 | 31.2 | 2.7 | 61.6 | 7.1 | 3.69 | 0.19 | 82.4 | 2.64 | 113.6 |
| CU-101 | Metamorphic | 22.0526 | −80.2922 | 9.7 | 81 | 1254 | 67.7 | 5.6 | 73.0 | 8.2 | 6.46 | 0.27 | 90.5 | 1.34 | 158.2 |
| CU-102 | Sedimentary | 22.3011 | −80.5004 | 1.0 | 19 | 1327 | 30.3 | 3.0 | 32.2 | 5.3 | 6.41 | 0.84 | 39.6 | 1.31 | 69.9 |
| CU-104 | Sedimentary | 22.7587 | −80.3621 | 0.4 | 127 | 1120 | ND | ND | ND | ND | ND | ND | 58.9 | ND | ND |
| CU-106 | Igneous | 22.7068 | −80.3667 | 0.5 | 133 | 1272 | 4.8 | 0.5 | 9.0 | 1.2 | 3.80 | 0.15 | 27.4 | 5.65 | 32.2 |
| CU-107 | Undetermined | 22.5354 | −79.8796 | 1.1 | 37 | 1250 | 22.3 | 2.1 | 29.2 | 4.6 | 5.30 | 0.60 | 75.7 | 3.39 | 98.0 |
| CU-108 | Igneous | 22.3924 | −79.6691 | 2.0 | 66 | 1370 | 76.5 | 7.7 | 79.0 | 18.5 | 6.74 | 1.44 | 111.7 | 1.46 | 188.2 |
| CU-109 | Igneous | 22.3570 | −79.7612 | 2.3 | 68 | 1373 | 96.1 | 9.3 | 98.5 | 12.9 | 6.81 | 0.64 | 155.0 | 1.61 | 251.1 |
| CU-110 | Metamorphic | 21.9187 | −80.2659 | 10.5 | 76 | 1029 | 46.6 | 4.0 | 57.5 | 6.5 | 5.66 | 0.26 | 119.6 | 2.57 | 166.2 |
| CU-111 | Metamorphic | 22.0895 | −79.9169 | 8.2 | 17 | 1489 | 189.0 | 17.7 | 193.0 | 23.4 | 6.91 | 0.54 | 108.7 | 0.58 | 297.7 |
| CU-112 | Metamorphic | 21.8326 | −80.1503 | 9.4 | 71 | 1059 | 42.0 | 3.7 | 46.4 | 5.3 | 6.26 | 0.31 | ND | ND | ND |
| CU-113 | Metamorphic | 21.8376 | −80.1045 | 11.7 | 56 | 1059 | 43.3 | 3.6 | 48.3 | 5.4 | 6.23 | 0.24 | 104.9 | 2.42 | 148.2 |
| CU-114 | Metamorphic | 22.1056 | −80.2253 | 8.9 | 32 | 1395 | 68.3 | 5.8 | 67.9 | 7.9 | 6.99 | 0.38 | 84.5 | 1.24 | 152.8 |
| CU-115 | Metamorphic | 22.1106 | −80.1291 | 10.9 | 333 | 1328 | 106.0 | 9.4 | 86.5 | 9.9 | 8.42 | 0.50 | 96.8 | 0.91 | 202.8 |
| CU-116 | Metamorphic | 22.0277 | −79.9889 | 9.9 | 40 | 1440 | 72.4 | 6.0 | 76.6 | 9.2 | 6.59 | 0.39 | 114.6 | 1.58 | 187.0 |
| CU-117 | Metamorphic | 22.0494 | −79.8431 | 6.5 | 40 | 1489 | 145.0 | 14.1 | 178.0 | 24.8 | 5.80 | 0.62 | 102.4 | 0.71 | 247.4 |
| CU-118 | Igneous | 22.3751 | −79.8175 | 4.6 | 42 | 1428 | 94.3 | 8.8 | 139.0 | 28.9 | 4.89 | 0.90 | 61.5 | 0.65 | 155.8 |
| CU-119 | Sedimentary | 22.5668 | −80.2220 | 1.6 | 707 | 1361 | 14.5 | 1.2 | 16.6 | 2.0 | 5.87 | 0.27 | 97.9 | 6.75 | 112.4 |
| CU-120 | Sedimentary | 22.4431 | −80.4809 | 0.7 | 54 | 1313 | 9.8 | 0.9 | 14.7 | 1.8 | 4.63 | 0.18 | 142.7 | 14.58 | 152.5 |
| CU-121 | Sedimentary | 22.4442 | −80.4448 | 0.6 | 49 | 1300 | 6.5 | 0.6 | 10.1 | 1.3 | 4.42 | 0.15 | 108.7 | 16.69 | 115.2 |
| CU-122 | Sedimentary | 22.5047 | −80.2907 | 0.5 | 2 | 1285 | 6.0 | 4.9 | 7.9 | 1.0 | 5.06 | 3.58 | 175.7 | 29.08 | 181.7 |
| CU-124 | Sedimentary | 22.7587 | −80.3621 | 0.5 | 19 | 1100 | ND | ND | ND | ND | ND | ND | 57.8 | ND | ND |
| CU-131 | Igneous | 22.3547 | −80.5088 | 0.6 | 172 | 1326 | 5.9 | 0.5 | 9.0 | 1.2 | 4.48 | 0.19 | 10.0 | 1.70 | 15.8 |
| CU-132 | Sedimentary | 22.4918 | −80.2963 | 0.5 | 23 | 1329 | 3.4 | 3.0 | 4.8 | 0.7 | 4.65 | 3.29 | 75.4 | 22.12 | 78.8 |

ND: no data.

**Table 2. (a)** Correlation coefficients for linear regressions. **(b)** $P$ values for linear regressions. TS2

(a)

| 10Be-derived erosion | Ratio of rock dissolution to 10Be-derived erosion | 26Al/10Be ratio | Quartz yield | Carbonate dissolution rate | Silicate dissolution rate | Evaporite dissolution rate | Sum of rock dissolution and 10Be-derived | Mean basin slope | Total basin area | Mean annual precipitation | % Agricultural land | Mean basin elevation | |
|---|---|---|---|---|---|---|---|---|---|---|---|---|---|
| 0.12 | 0.36 | 0.25 | 0.32 | 0.59 | 0.32 | 0.65 | 0.65 | 0.19 | -0.05 | 0.07 | -0.27 | 0.22 | Rock dissolution rate |
| | -0.55 | 0.64 | -0.01 | 0.41 | 0.42 | -0.32 | 0.83 | 0.44 | 0.18 | 0.47 | -0.34 | 0.34 | 10Be-derived erosion |
| | | -0.46 | 0.45 | -0.43 | -0.02 | 0.85 | -0.22 | -0.47 | -0.20 | -0.18 | 0.16 | -0.45 | Ratio of rock dissolution to 10Be-derived erosion |
| | | | 0.17 | 0.60 | 0.14 | -0.29 | 0.64 | 0.58 | 0.26 | 0.05 | -0.49 | 0.56 | 26Al/10Be ratio |
| | | | | 0.13 | -0.11 | 0.32 | 0.17 | 0.39 | -0.19 | -0.12 | -0.61 | 0.33 | yield |
| | | | | | 0.16 | -0.22 | 0.64 | 0.68 | -0.02 | -0.10 | -0.54 | 0.74 | Carbonate dissolution rate |
| | | | | | | 0.04 | 0.46 | -0.13 | 0.23 | 0.52 | 0.24 | -0.12 | Silicate dissolution rate |
| | | | | | | | 0.11 | -0.36 | -0.09 | 0.07 | 0.10 | -0.38 | Evaporite dissolution rate |
| | | | | | | | | 0.44 | 0.09 | 0.33 | -0.40 | 0.41 | Sum of rock dissolution and 10Be- |
| | | | | | | | | | -0.11 | -0.14 | -0.79 | 0.95 | Mean basin |
| | | | | | | | | | | 0.23 | 0.02 | -0.05 | Total basin area |
| | | | | | | | | | | | 0.09 | -0.27 | Mean annual precipitation |
| | | | | | | | | | | | | -0.74 | % Agricultural land |

the highest [10]Be concentrations (Fig. 5) were collected in the northwestern part of the field area in basins predominately underlain by sedimentary rocks (Fig. 2). For the most part, samples from basins dominantly underlain by igneous and metamorphic rocks plot to the left on the two-isotope diagram and have higher $^{26}$Al / $^{10}$Be ratios than quartz from basins underlain by sedimentary rocks. Basins draining primarily sedimentary lithologies have the highest ratio of rock dissolution to $^{10}$Be-derived erosion rates. Seven basins (CU-106, CU-119, CU-120, CU-121, CU-122, CU-131, and CU-132) stand out from the rest (Fig. 5) and are clustered in the northwestern part of our field area. These basins have much lower than average $^{10}$Be-derived erosion rates (3.4–14.5 Mg km$^{-2}$ yr$^{-1}$), low $^{26}$Al / $^{10}$Be ratios (3.80–5.87), and rock dissolution rates 1.7–29 times higher than the $^{10}$Be-derived rates of erosion. All but CU-131 are underlain primarily by sedimentary rocks.

Neon isotope measurements (Table S10) revealed high total neon concentrations with isotope composition indistinguishable from atmosphere, so excess $^{21}$Ne was indistinguishable from zero and could not be quantified. Expected cosmogenic $^{21}$Ne concentrations in the samples we analyzed, calculated from observed $^{10}$Be concentrations and the assumption of steady erosion (3–6 M atoms g$^{-1}$ cosmogenic $^{21}$Ne), would comprise less than 2 % of the total amount of $^{21}$Ne we observed and would not be detectable at typical measurement uncertainties. The neon isotope measurements are not inconsistent with the $^{26}$Al and $^{10}$Be data but provide no additional information.

## 6 Discussion

In central Cuba, erosion rates inferred from the concentration of $^{10}$Be in river sand vary by more than an order of magnitude. The lowest $^{10}$Be-inferred erosion rate (3.4 Mg km$^{-2}$ yr$^{-1}$; 1.3 m Myr$^{-1}$) is less than those measured in tectonically stable arid landscapes including Namibia and Australia (Bierman and Caffee, 2001; Codilean et al., 2021). The highest $^{10}$Be-inferred rate (189 Mg km$^{-2}$ yr$^{-1}$; 73 m Myr$^{-1}$) exceeds those measured in temperate, humid, tectonically stable areas, such as the southern Appalachian Mountains (Portenga et al., 2019; Duxbury et al., 2015; Linari et al., 2017), and is similar to or less than rates measured on other Caribbean islands includ-

(b)

| 10Be-derived erosion | Ratio of rock dissolution to 10Be-derived erosion | 26Al/10Be ratio | Quartz yield | Carbonate dissolution rate | Silicate dissolution rate | Evaporite dissolution rate | Sum of rock dissolution and 10Be-derived erosion rate | Mean basin slope | Total basin area | Mean annual precipitation | % Agricultural land | Mean basin elevation | |
|---|---|---|---|---|---|---|---|---|---|---|---|---|---|
| 0.583 | 0.090 | 0.260 | 0.147 | 0.002 | 0.124 | 0.000 | 0.001 | 0.363 | 0.816 | 0.753 | 0.184 | 0.296 | Rock dissolution rate |
| | 0.006 | 0.001 | 0.977 | 0.054 | 0.047 | 0.143 | 0.000 | 0.029 | 0.397 | 0.020 | 0.113 | 0.105 | 10Be-derived erosion |
| | | 0.028 | 0.034 | 0.039 | 0.941 | 0.000 | 0.315 | 0.024 | 0.369 | 0.421 | 0.467 | 0.031 | Ratio of rock dissolution to 10Be-derived erosion |
| | | | 0.443 | 0.002 | 0.530 | 0.184 | 0.001 | 0.003 | 0.222 | 0.818 | 0.017 | 0.004 | 26Al/10Be ratio |
| | | | | 0.563 | 0.629 | 0.141 | 0.442 | 0.064 | 0.393 | 0.581 | 0.002 | 0.118 | Quartz yield |
| | | | | | 0.456 | 0.291 | 0.001 | 0.000 | 0.941 | 0.626 | 0.006 | 0.000 | Carbonate dissolution rate |
| | | | | | | 0.848 | 0.025 | 0.540 | 0.279 | 0.007 | 0.248 | 0.571 | Silicate dissolution rate |
| | | | | | | | 0.605 | 0.076 | 0.653 | 0.754 | 0.636 | 0.062 | Evaporite dissolution rate |
| | | | | | | | | 0.037 | 0.683 | 0.120 | 0.056 | 0.051 | Sum of rock dissolution and 10Be-derived erosion |
| | | | | | | | | | 0.596 | 0.501 | 0.000 | 0.000 | Mean basin slope |
| | p<0.01 | | | | | | | | | 0.265 | 0.936 | 0.817 | Total basin area |
| | p<0.05 | | | | | | | | | | 0.679 | 0.175 | Mean annual precipitation |
| | p<0.1 | | | | | | | | | | | 0.000 | % Agricultural land |

ing Puerto Rico and Dominica (Quock et al., 2021; Brocard et al., 2015; Brown et al., 1995).

Variability in $^{10}$Be concentration, and thus inferred rates of erosion, between central Cuban drainage basins, many within just a few tens of kilometers of each other with similar basin slope, suggests significant landscape-scale controls on $^{10}$Be concentration and thus mass loss. Indeed, we find that $^{10}$Be-determined erosion rates are positively correlated with slope ($R^2 = 0.20$, $p = 0.03$), mean annual precipitation ($R^2 = 0.22$, $p = 0.02$), and rates of silicate dissolution ($R^2 = 0.17$, $p = 0.05$) (Fig. 6, Table 2). Erosion rates are lowest for basins underlain dominantly by sedimentary rocks and highest for basins underlain by metamorphic rocks (Fig. 3). However, accurately quantifying rates of denudation (total mass loss) from central Cuban landscapes is complicated by significant export of mass in solution and near-surface quartz enrichment. In the sections that follow, we discuss the $^{10}$Be data in the context of dissolved load export in river water and the landscape-scale insight on active processes provided by dual-isotope measurements ($^{26}$Al and $^{10}$Be) made in riverine quartz.

## 6.1 Cosmogenic erosion rates underestimate landscape-scale mass loss in Cuba

Our data clearly show that significant, landscape-scale mass loss is occurring by solution in central Cuba. Rock dissolution rates exceed, some by more than an order of magnitude, corresponding $^{10}$Be-derived mass loss rates in central Cuba, demonstrating that the cosmogenic nuclide measurements are an incomplete assessment of total mass loss from the landscape. Rock dissolution rates are greater than cosmogenic erosion rates for 18 of the 23 basins we analyzed, and the median rock dissolution rate in Cuba is 2.15 times higher than the median cosmogenic-nuclide-derived rate (Table 1, Fig. 7).

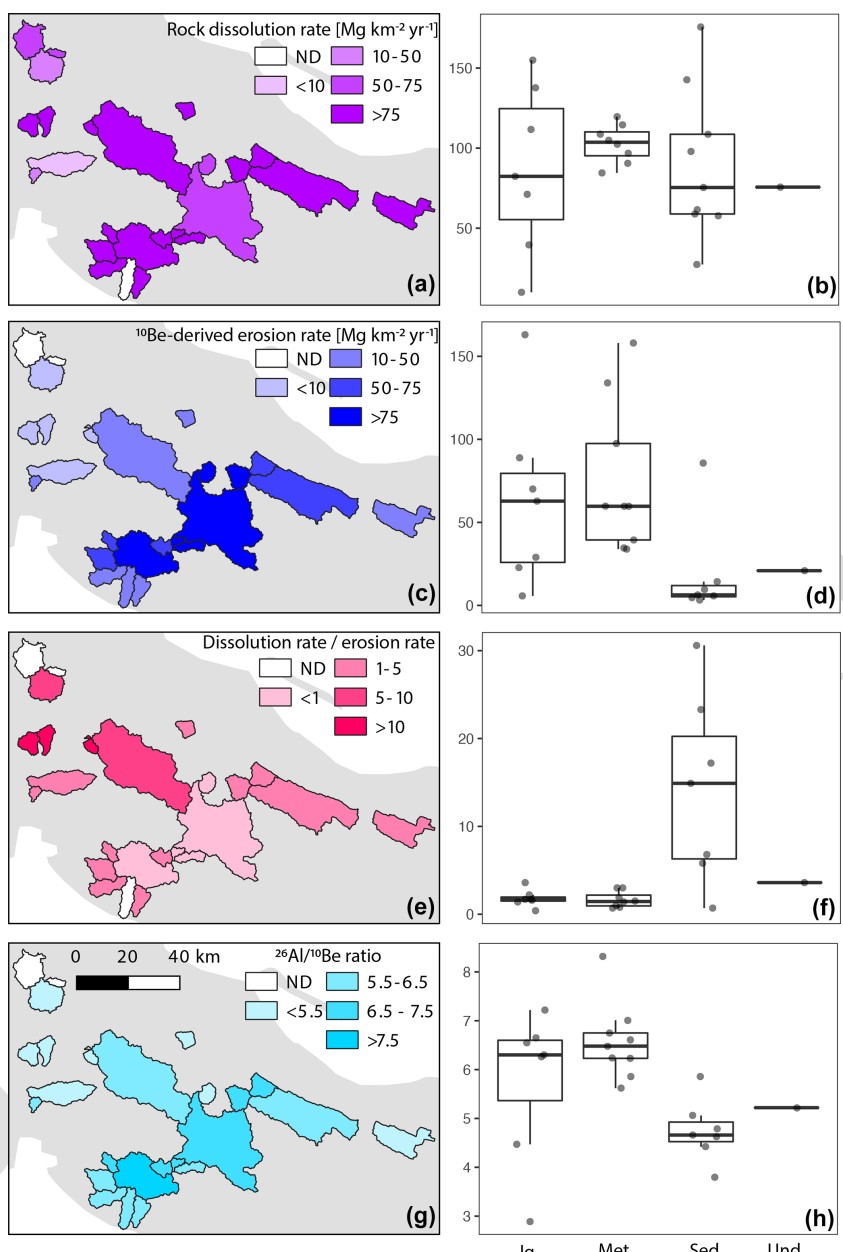

**Figure 3.** Maps showing rates of landscape change and isotopic data for each study watershed. **(a, b)** Rock dissolution rates. **(c, d)** [10]Be-derived erosion rates. In both maps, darker colors in the basins indicate faster rates of landscape change. **(e, f)** Ratio between rock dissolution rates and [10]Be-derived erosion rates; darker colors indicate larger ratios. **(g, h)** [26]Al / [10]Be ratio. Darker colors are higher ratios. Box plots show the maximum and minimum values in the lines extending from the box; the upper side of the box represents the upper quartile, the line inside the box represents the median value, and the bottom of the box represents the lower quartile. *Y*-axis units are the same as shown in the corresponding map legend.

Although rock dissolution rates and cosmogenic-nuclide-derived erosion rates integrate over different timescales, they have been compared in other areas. Rock dissolution rates in our study represent a single sample for each watershed integrated with annual discharge rates, although weathering fluxes must respond to landscape and hydrologic conditions over centuries to millennia as soil and regolith develop. Cosmogenic-nuclide-derived rates integrate over the time it takes to remove $\sim 2\,\mathrm{m}$ of material from the surface; in our field area, this represents many tens to a few hundred thousand years at most. In general, higher rock dissolution rates are favored by higher temperatures, higher precipitation, and longer mineral residence times in the shallow subsurface, which are all more likely to be found in low-relief

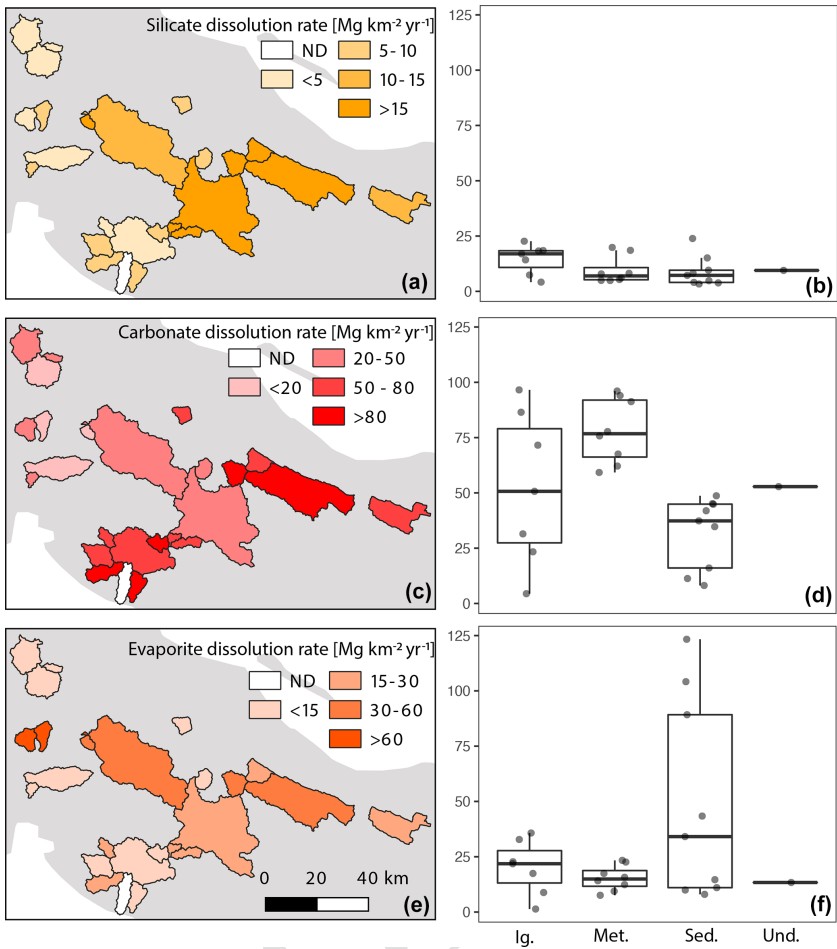

**Figure 4.** Dissolution rates partitioned by lithology. **(a, b)** Silicate. **(c, d)** Carbonate. **(e, f)** Evaporite. Darker colors represent higher rates. Box plots show the maximum and minimum values in the lines extending from the box; the upper side of the box represents the upper quartile, the line inside the box represents the median value, and the bottom of the box represents the lower quartile.

regions of the tropics and less likely to be found in higher-relief, commonly glaciated, temperate and polar regions.

Rock dissolution rates that significantly exceed corresponding [10]Be-inferred rates have also been reported from Uganda (Hinderer et al., 2013) and Cameroon (Regard et al., 2016), where they were attributed to the influence of easily weathered volcanic tephra and deep weathering associated with thick regolith, respectively. Most other studies that compare rock dissolution rates and [10]Be-derived erosion rates in the tropics documented rock dissolution rates within the range of cosmogenic-nuclide-derived rates (von Blancken-burg et al., 2004; Salgado et al., 2006; Cherem et al., 2012; Sosa Gonzalez et al., 2016b; Quock et al., 2021). Cuba is different.

The discordance between high rock dissolution rates and low [10]Be-derived erosion rates in central Cuba suggests that significant rock weathering is occurring below the depth of most cosmogenic nuclide production (Bierman and Steig, 1996; Fig. 1). The discordance, along with high rates of car-

bonate and evaporite dissolution in some basins, suggests that many lithologies in our field area are highly suscepti-ble to dissolution. Bierman et al. (2020) attribute high rock dissolution rates and the relationship between stream water chemistry and bedrock type in central Cuba to extensive rock–groundwater interaction along subsurface flow paths, controlled by ongoing bedrock uplift and associated rock fracturing. The prevalence of rock dissolution at depth in Cuba is consistent with findings from other humid, tropical landscapes, including Puerto Rico (White et al., 1998; Kurtz et al., 2011; Chapela Lara et al., 2017; Moore et al., 2019), Guadeloupe, Martinique, Dominica (Rad et al., 2007), and Hawaii (Schopka and Derry, 2012).

We observed no correlation between [10]Be-derived erosion and rock dissolution rates in central Cuba (Fig. 7), in con-trast to other studies in the tropics that have observed gen-erally positive correlations (Salgado et al., 2006; Cherem et al., 2012; Sosa Gonzalez et al., 2016b). The lack of correla-tion suggests that mass loss below several meters, the depth

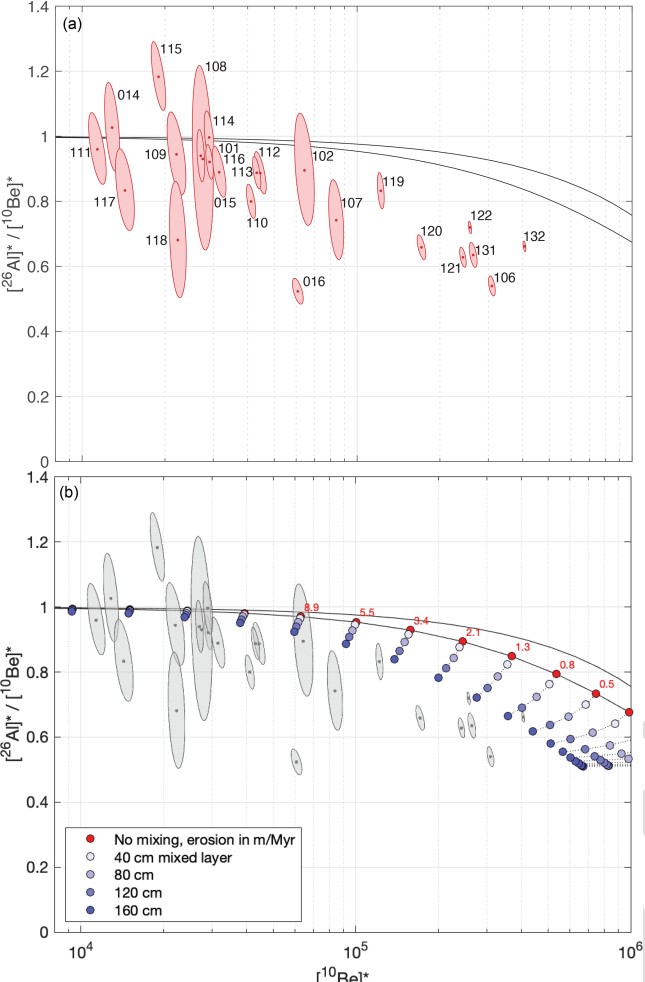

at which most cosmogenic nuclides are produced, is an important component of denudation in Cuba. Discordance between high rock dissolution rates and low $^{10}$Be-derived erosion rates observed in Cuba occurs in basins with different underlying lithologies (Fig. 6). Such widespread discordance suggests that deep chemical weathering is occurring throughout central Cuba.

Carbonate weathering dominates river water geochemistry in central Cuba. Our analysis of Cuban water composition suggests that the rate of carbonate dissolution varies widely and in most basins we sampled exceeds by several-fold the rate of silicate dissolution (Fig. 4). Silicate dissolution rates are low ($< 25\,\mathrm{Mg\,km^{-2}\,yr^{-1}}$) and similar between all lithologies. Export rates of elements calculated to reflect the presence of evaporite minerals are also generally low ($< 35\,\mathrm{Mg\,km^{-2}\,yr^{-1}}$), except in four basins dominated by sedimentary rocks (CU-120, CU-121, CU-122, CU-132). Water geochemistry data from four of these basins suggest the presence of significant evaporite deposits due to high concentrations of Cl, $SO_4$, Br, and Na (Bierman et al., 2020). Together these data imply that lithologies underlying the basins we sampled are not uniform and that silicate rocks do not account for most of the dissolved mass loss in at least some, and likely many, of the basins we sampled.

Together, underlying lithology and topography are important controlling factors in how and how rapidly the Cuban landscapes we studied are losing mass by both physical and chemical weathering. Lowland basins, primarily underlain by sedimentary rocks, on average have low rates of $^{10}$Be-inferred mass loss and high rates of dissolution. Six basins underlain by sedimentary lithologies (CU-106, CU-119, CU-120, CU-121, CU-122, and CU-132) have the highest $^{10}$Be concentrations and lowest erosion rates, indicating near-surface residence times several to more than 10 times longer for the quartz we analyzed from these basins than from other basins. All are low-slope (0.5 to 1.6°). These six basins also demonstrate the greatest disparity between high rock dissolution rates and low $^{10}$Be-derived erosion rates (5.7-29X). One basin underlain by igneous rocks (CU-131) has a similarly low slope (0.6°) and high $^{10}$Be concentration but a much lower ratio of dissolution to erosion rates (1.7), likely reflecting the paucity of readily soluble minerals. As a result, $^{10}$Be-derived erosion rates are weakly and positively correlated with average basin slope ($R^2 = 0.20$, $p = 0.03$), but rock dissolution rates are not correlated ($R^2 = 0.04$, $p = 0.36$) with slope.

## 6.2 Low $^{26}$Al / $^{10}$Be ratio evidence for a deep mixed surface layer and possible quartz enrichment

The $^{26}$Al / $^{10}$Be ratios suggest that most sediment we collected from central Cuban rivers does not have a simple exposure history. $^{26}$Al / $^{10}$Be data in 16 of 24 sampled basins are inconsistent with steady surface erosion (Fig. 5). Many of the basins with the lowest $^{26}$Al / $^{10}$Be ratios drain predomi-

**Figure 5.** $^{26}$Al / $^{10}$Be two-isotope plots. To permit comparison of data from different locations and elevations on the same plot, nuclide concentrations have been normalized by dividing measured concentrations by calculated mean production rates in the respective drainage basins using production rate calculations from version 3 of the online exposure age calculator described by Balco et al. (2008) and subsequently updated. In both plots, uncertainty ellipses denote 68 % confidence regions for the normalized nuclide concentrations, and the black lines are the boundaries of the simple exposure region (Lal, 1991) calculated using the conventional assumption of steady block erosion without vertical mixing. Panel (**a**) shows that $^{26}$Al / $^{10}$Be ratios from basins with high nuclide concentrations, implying low erosion rates, are systematically lower than predicted by steady erosion without vertical mixing. Panel (**b**) shows that this inconsistency can, at least in part, be explained by the presence of a mixed layer of at least 160 cm. Circles show expected steady-state nuclide concentrations in a fully mixed surface layer calculated according to Lal and Chen (2005) for a range of erosion rates and mixed layer thicknesses, which highlights that sediment derived from a deep mixed layer has lower nuclide concentrations and lower $^{26}$Al / $^{10}$Be ratios than would be expected if the mixed layer were absent. The sample ID is as in panel (**a**).

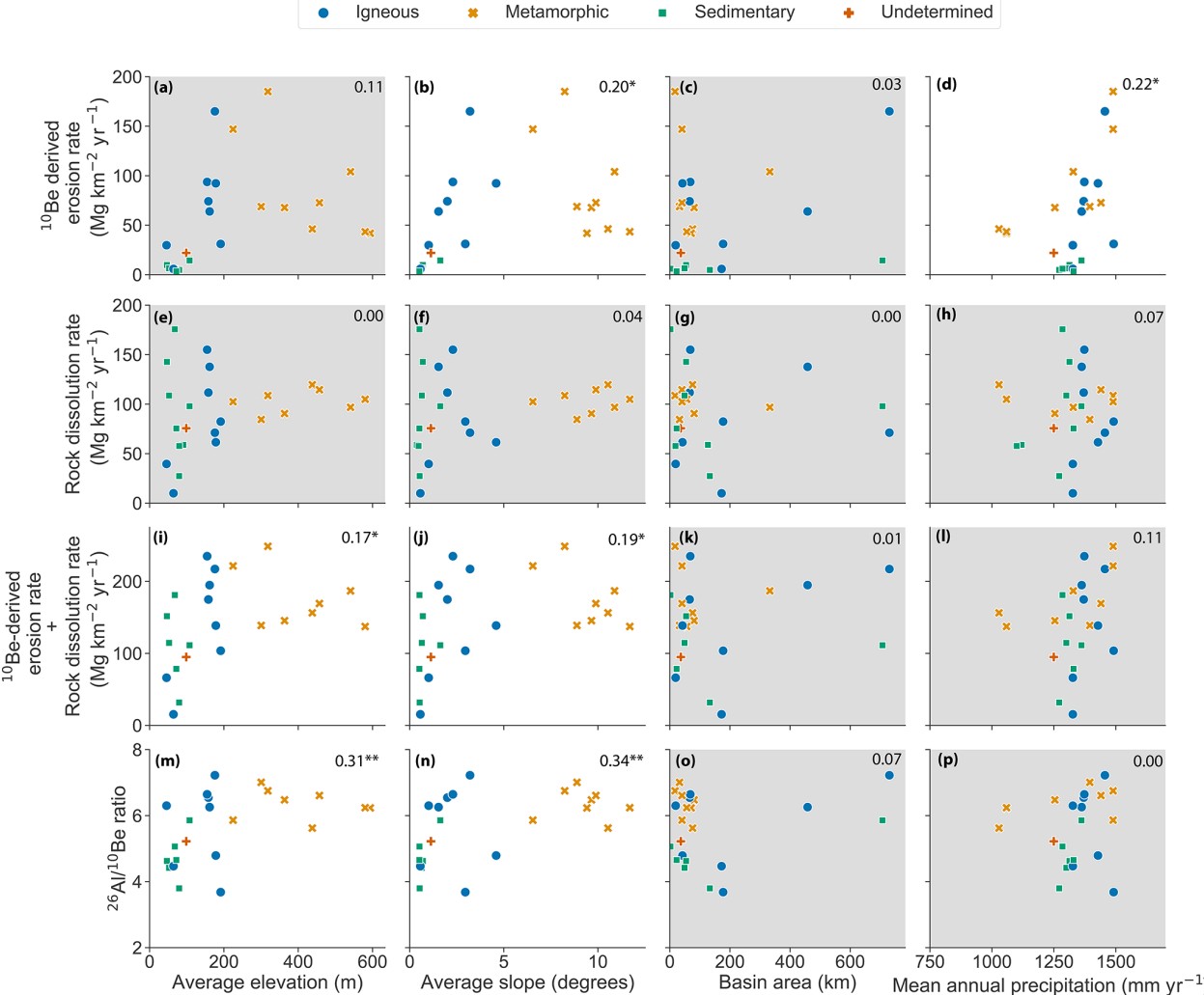

**Figure 6.** Relationship of measured $^{10}$Be-derived erosion rates, chemical denudation rates, and the sum of chemical denudation rates and $^{10}$Be-derived erosion rates to basin characteristics. Differently shaped and colored points represent the dominant underlying rock type in that basin. Plots with $p > 0.05$ are shown with a gray background. Small numbers in the upper right are $R^2$ values; * indicates $p \leq 0.05$, ** $p < 0.01$.

nantly marine sedimentary lithologies and have low average basin slopes (0.5–0.7°); the remaining basin drains primarily igneous rocks and has an average basin slope of 0.6°. These are the same seven basins discussed in the section above, all but one of which have high ratios of dissolved load to $^{10}$Be-inferred erosion rates. There is a significant positive ($R^2 = 0.34$, $p = 0.003$) relationship between average basin slope and $^{26}$Al / $^{10}$Be.

Observed $^{26}$Al / $^{10}$Be ratios in some of the low-ratio samples are consistent with bioturbation and prolonged near-surface exposure (Struck et al., 2018). We suspect that at least some of the inconsistency between measured $^{26}$Al / $^{10}$Be ratios and those predicted by a simple, steady-state surface erosion model is due to (deep) soil mixing. Typically, the lower boundary of the simple exposure region of a two-isotope diagram (Fig. 5) is constructed based on the assumption that

all grains move monotonically towards the surface at the rate that the surface is eroding (Granger, 2006). Vertical mixing, due to bioturbation or other soil processes taking place in the upper layers of soil, violates this assumption. Within a mixed soil layer, grains circulate at a higher velocity than the erosion rate and therefore experience an average production rate lower than the surface rate and spend time below the surface where the rate of nuclide decay may exceed the rate of nuclide production. During burial, $^{26}$Al / $^{10}$Be ratios decrease and diverge from those predicted by the steady-state surface erosion model (Fig. 5).

Rapid chemical mass loss due to the presence of readily soluble evaporite and marine or igneous lithologies in some basins likely enriches the remaining sediment in quartz. The combination of mass loss by rapid rock dissolution and the retention of weathering residuum favored by subdued topog-

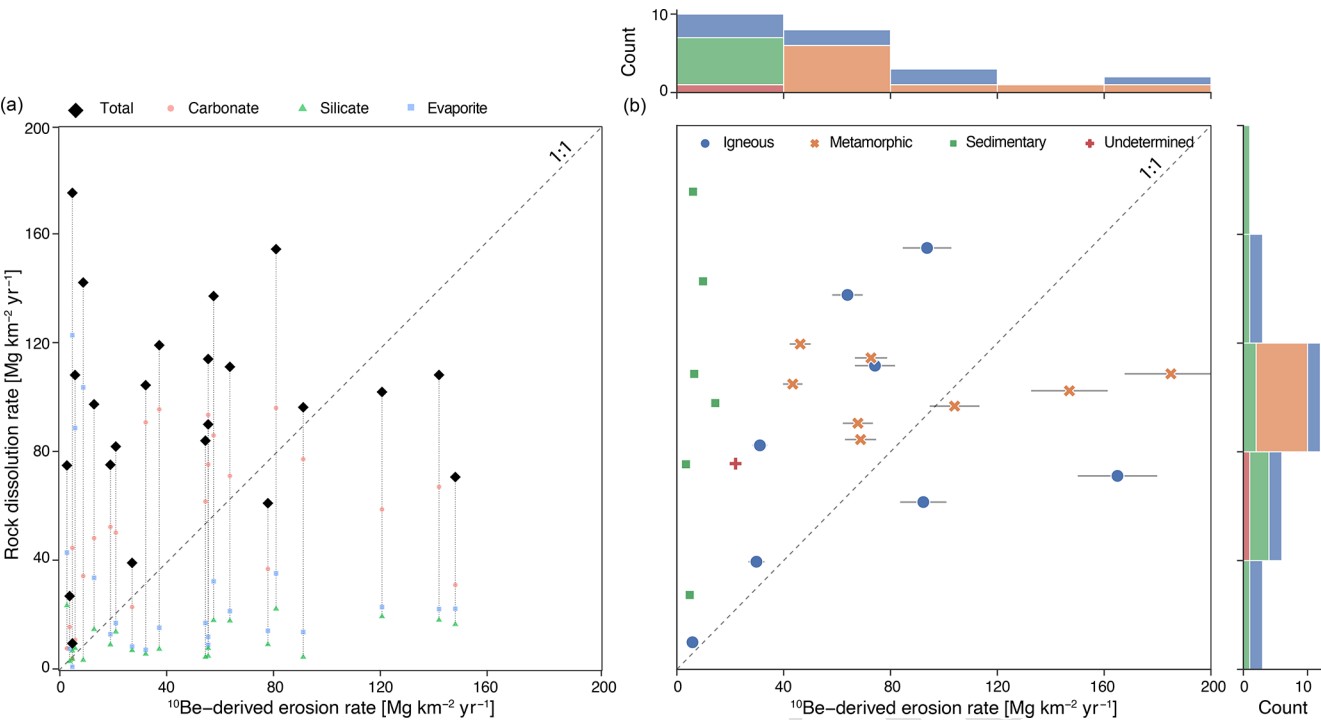

**Figure 7.** Scatterplot of rock dissolution rates vs. [10]Be-derived erosion rates. **(a)** Data shown for calculations of carbonate, silicate, and evaporite dissolution rates. **(b)** Data shown for dominant lithologies underlying each sampled basin. Horizontal lines extending from the points demonstrate the uncertainty associated with the calculation of [10]Be-derived erosion rates. Histograms on axes show the distribution of data. The dashed line is 1 : 1.

raphy in low-relief basins allows less-soluble material (e.g., quartz) to accumulate at and near the surface, creating thick regolith. Extensive vertical mixing of near-surface soil, as is expected for flat, forested landscapes where the rate of bioturbation is likely high in relation to slow erosion rates, leads to longer residence times for these residual mineral grains and therefore a lower $^{26}\text{Al}/^{10}\text{Be}$ ratio in a mixed surface layer compared to a surface eroding at the same rate without vertical mixing.

This assertion is supported by the consistency between measured $^{26}\text{Al}/^{10}\text{Be}$ ratios, expected nuclide concentrations, and ratios calculated assuming the presence of a mixed surface layer (per Lal and Chen, 2005, Eq. 12). Expected $^{26}\text{Al}/^{10}\text{Be}$ ratios calculated assuming a mixed layer depth of 40–160 cm agree well with measured low $^{26}\text{Al}/^{10}\text{Be}$ ratios from basins CU-119, CU-122, and CU-132 (Fig. 5). This mixed layer depth range is consistent with the soil depths of 90–150 cm reported for the location of these basins (Bennett and Allison, 1928). In deeply weathered tropical soils, bioturbation can extend to depths of several meters (von Blanckenburg et al., 2004), so it is plausible that mixing depths are even greater than the model suggests, providing an explanation for the $^{26}\text{Al}/^{10}\text{Be}$ ratios measured in CU-120, CU-121, CU-131, and CU-106. We were not able to measure regolith depths in the drainage basins we sampled.

The $^{26}\text{Al}/^{10}\text{Be}$ ratio in other samples (e.g., CU-106, CU-118, and CU-110) is too low to be attributed solely to the effects of a deep mixed surface layer and requires some fraction of the sample to have experienced both surface exposure and a significant period of burial well below the surface where cosmogenic nuclide production is negligible. Factors that could lead to this low ratio include the incorporation of previously deeply buried sediment through channel avulsion (Wittmann et al., 2011) or incision into terraces (Hu et al., 2011). We conclude that terrace storage, along with a combination of quartz enrichment due to high chemical weathering rates of soluble marine rocks in combination with very low-slope basins and a deep mixing layer, generates detrital quartz with high concentrations of $^{10}\text{Be}$ and lower than expected $^{26}\text{Al}/^{10}\text{Be}$ ratios.

## 6.3 Constraining total rates of landscape denudation

The disagreement between high rock dissolution rates and low $^{10}\text{Be}$-derived erosion rates raises questions about how to best characterize total landscape denudation rates. It is clear from our dataset that neither cosmogenic nuclide measurements nor stream solute fluxes capture all or even, in some cases, the majority of landscape denudation in central Cuba. Evidence for deep rock dissolution presented in Sect. 6.1 suggests that sediments and solutes are being sourced at

least partially from different depths in the landscape. Because most mass loss in much of central Cuba occurs in solution (rock dissolution rates are higher than [10]Be-derived erosion rates in 18 of 23 basins), rock dissolution rates typically represent a minimum bound on total landscape denudation.

One complication with directly comparing [10]Be-derived erosion rates and rock dissolution rates is that they integrate over different timescales. Our rock dissolution rates are based on water samples collected once during the rainy season. The dissolved load of those samples was scaled using modeled mean annual discharge and the assumption that the concentration of each species is discharge-independent. Thus, we just have a single point snapshot of rock dissolution rates. In contrast, the [10]Be-derived erosion rates are integrated over the time it takes the quartz currently in the river channel to move from $\sim 2$ m below the surface to the surface. The high nuclide concentrations we measured in Cuba suggest that tens of thousands to perhaps a few hundred thousand years are integrated into these measurements. Thus, the comparison we (and others before us) make between rock dissolution rates and [10]Be-derived erosion rates implicitly assumes that the two measurements are steady enough through time to be compared.

Treating the removal of mass in solution and through physical erosion as entirely discrete processes happening at different depths in the landscape sets an upper limit on total landscape denudation: the sum of inferred rock dissolution rates and [10]Be-derived erosion rates. Summing [10]Be-derived erosion rates and chemical denudation rates increases estimates of total landscape denudation across study basins by a factor of 1.4–30 (mean 6.3, median 2.7) above [10]Be-derived erosion rates. Disregarding the six basins with evidence of evaporite deposits (CU-106, CU-119, CU-120, CU-121, CU-122, and CU-132) leads to an average increase of a factor of 2.6 (median 2.5) above [10]Be-derived erosion rates. These mean and median values for the basins without evaporites are between the reported CEF of 1.79 for the Luquillo Critical Zone Observatory in humid, tropical Puerto Rico (Riebe and Granger, 2013) and the CEF of 3.2 for the thickly saprolite-mantled, tropical environment of southern Cameroon (Regard et al., 2016). These comparisons suggest that for landscapes with a significant proportion of total denudation occurring through deep rock dissolution, summing rock dissolution rates and cosmogenic-nuclide-derived rates provides a reasonable estimate of total landscape denudation.

In landscapes like central Cuba, total denudation rates may be difficult to predict based on landscape metrics. Summed chemical denudation rates and cosmogenic-nuclide-derived erosion rates are not correlated with rock type, as rock type appears to have opposing influences on these rates (i.e., basins underlain by sedimentary rocks had the highest rock dissolution rates but the lowest cosmogenic-nuclide-derived rates, Fig. 3). Summed rock dissolution rates and cosmogenic-nuclide-derived rates do increase with mean basin slope ($R^2 = 0.19$, $p = 0.04$) and mean basin elevation

($R^2 = 0.17$, $p = 0.05$) (Fig. 5), but those relationships are confounded because [10]Be-derived rates are highest in high-elevation, steep basins and rock dissolution rates are highest in low-slope, low-elevation basins – relationships that are primarily controlled by the influence of rock type on these two different mass loss processes.

In central Cuba, the lack of correlation between rock dissolution rates and [10]Be-derived erosion rates (Fig. 7) suggests a possible mechanism for limiting total reduction in landscape relief. While global data demonstrate significant, positive correlations between [10]Be-derived erosion rates and basin slope and relief (Portenga and Bierman, 2011), accounting for the influence of rock dissolution may alter this dynamic. The possibility of combined physical and chemical processes limiting reductions in relief has significant implications for the study of deeply weathered, high-relief tropical landscapes. The dual importance of rock dissolution in low-lying areas and physical erosion in steeper terrain could explain the relationship behind sustained high-relief topography and low [10]Be-derived erosion rates common across some tropical landscapes, such as Brazil (Vasconcelos et al., 2019) or Sri Lanka (von Blanckenburg et al., 2004). Landscapes with high rock dissolution rates and low physical erosion rates appear to be relatively common (Larsen et al., 2014b). As lowlands weather primarily through rock dissolution and high-relief areas by physical erosion, total relief would change more slowly than [10]Be-estimated rate differentials would suggest.

Regardless of rock type, however, cosmogenic-nuclide-derived erosion rates are positively correlated with MAP ($R^2 = 0.22$, $p = 0.02$). While MAP does not vary widely across our study basins in central Cuba (956 to 1555 mm yr$^{-1}$), this correlation suggests a climatic control on denudation rates across this landscape. This finding is contrary to other studies in the humid tropics (von Blanckenburg et al., 2004) and beyond (Riebe et al., 2001b; Portenga and Bierman, 2011) that have found no correlation between climate variables and cosmogenic-nuclide-derived long-term erosion rates, but it is similar to recent findings in humid, temperate Tasmania (VanLandingham et al., 2022). Since in Cuba [10]Be-derived erosion rates are positively correlated with MAP but chemical denudation rates are uncorrelated with MAP, this trend likely highlights the importance of rainfall in allowing for the physical export of sediment from a drainage basin that is transport-limited rather than weathering-limited.

Our data clearly demonstrate that cosmogenic nuclide measurements can underestimate total denudation in landscapes with significant rock dissolution at depth, particularly in the tropics. This suggests that similar underestimates of total denudation rates produced by relying on measurements of cosmogenic nuclides may be a factor in other tropical landscapes. While rock dissolution rates in the tropics have been documented as among the highest globally (White and Blum, 1995; Rad et al., 2013; Larsen et al., 2014a),

a global compilation of [10]Be-derived erosion rates demonstrated that such isotopically determined rates of erosion are lower in the tropics than in all other climate zones, apart from arid regions (Portenga and Bierman, 2011). The contrast between these two depictions of tropical denudation suggests that [10]Be-derived erosion rates for tropical areas may be incomplete representations of total mass loss from these landscapes because dissolved loads are only partially (and perhaps minimally) accounted for by measurements of [10]Be in river sand. This discrepancy highlights the need for more studies that compare rock dissolution rates and cosmogenic-nuclide-derived rates to provide more accurate estimations of total landscape denudation (VanLandingham et al., 2022).

## 7 Conclusions

The first cosmogenic nuclide measurements from the island of Cuba provide insight into how mass is lost from landscapes in humid, tropical settings. Solution plays a large role in total mass flux, and significant mineral dissolution occurs along weathering fronts meters below the landscape surface. Rock type exerts the primary control on the pace of denudation, and precipitation influences rates of landscape change. We find evidence for thick, mixed surface layers in lowland basins, and river water chemistry data suggest that deep rock dissolution dominates denudational processes in low-slope basins where weathering products remain near the surface for long periods of time.

These findings highlight the necessity of accounting for mass loss by solution at depths below the penetration of most cosmic rays when interpreting cosmogenic-nuclide-derived erosion rates in landscapes with the potential for significant rock dissolution. The discrepancy between high rock dissolution rates and low [10]Be-derived erosion rates observed in central Cuba emphasizes how relying on cosmogenic nuclide measurements alone to determine total rates of mass loss from landscapes can lead to considerable underestimates of denudation. Summing mass loss rates in solution with mass loss rates inferred from cosmogenic nuclides provides an upper limit for total mass loss from landscapes when significant rock dissolution occurs below the penetration depth of cosmic-ray neutrons. These findings suggest that estimating rock dissolution rates is important when applying cosmogenic nuclides in landscapes, especially those which are humid, tropical, have soluble rocks, and/or have deep weathered regolith.

**Data availability.** The data are available at Pangaea (https://doi.org/10.1594/PANGAEA.940043 TS3, Campbell et al., 2022).

**Supplement.** The supplement related to this article is available online at: https://doi.org/10.5194/gchron-4-1-2022-supplement.

**Author contributions.** AHS, MKC, and PRB contributed to project design. AGA, AGM, AHS, HCA, MKC, PBR, and RSH conducted fieldwork. MKC prepared samples for laboratory analysis. GB, LBC, MC, AHS, PRB, AJH, and MKC contributed to cosmogenic nuclide analysis. MKC, PRB, and AHS led paper preparation; all authors assisted with data analysis as well as paper writing and review. GB, MKC, and AHS prepared the figures. AHS completed the chemical weathering calculations with support from DD.

**Competing interests.** At least one of the co-authors is a member of the editorial board of *Geochronology*. The peer-review process was guided by an independent editor, and the authors also have no other competing interests to declare.

**Disclaimer.** Publisher's note: Copernicus Publications remains neutral with regard to jurisdictional claims in published maps and institutional affiliations.

**Acknowledgements.** Support for fieldwork and analyses was provided by NSF EAR-1719249 and NSF EAR-1719240 to Paul R. Bierman and Amanda H. Schmidt, NSF EAR-1735676 to Paul R. Bierman, and Oberlin College funding to Amanda H. Schmidt. Researchers from Centro de Estudios Ambientales de Cienfuegos were supported by the MICATIN and ISOAGRI projects. We thank Jay Racela and Marika Massey-Bierman for their assistance with laboratory work and Marika Massey-Bierman, Monica Dix, Victor Manuel Fonseca Pérez, and Santiago Gil Pérez for assistance with fieldwork. We thank Erica Erlanger and Claire Lukens for constructive reviews. This work was prepared in part by LLNL under contract DE-AC52-07NA27344 (LLNL-JRNL-824414).

**Financial support.** This research has been supported by the National Science Foundation, Directorate for Geosciences (grant nos. EAR-1719249, EAR-1719240, and EAR-1735676), Oberlin College, and the Centro de Estudios Ambientales de Cienfuegos (MICATIN and ISOAGRI projects).

**Review statement.** This paper was edited by Hella Wittmann-Oelze and reviewed by Erica Erlanger and Claire E Lukens.

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

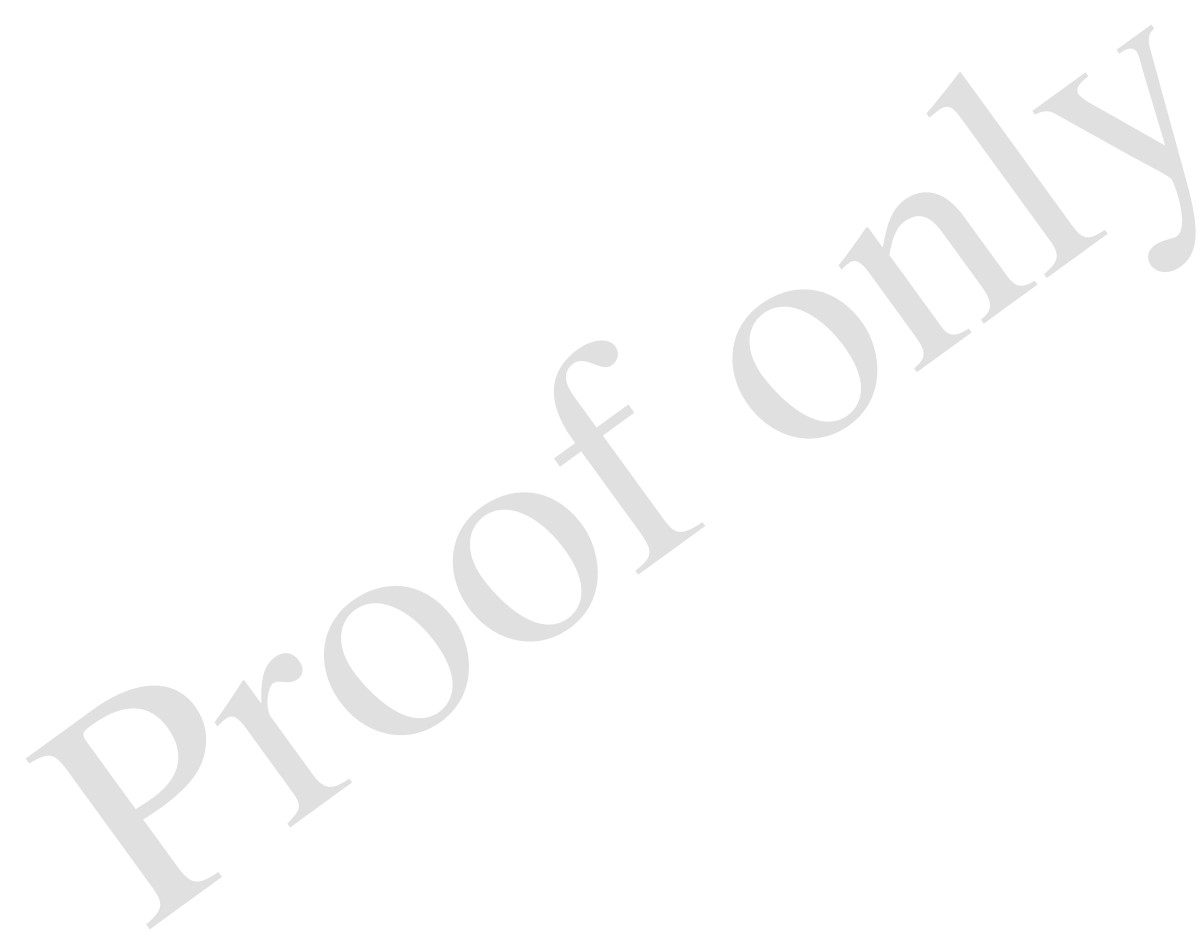

## Remarks from the typesetter