# Peer review of "Cosmogenic nuclide and solute flux data from central Cuban rivers emphasize the importance of both physical and chemical mass loss from tropical landscapes"

_Geochronology, 2021_

## Author Comment (AC1)

Our responses to the reviews are interspersed below. Our comments are in blue while the original text of the reviews is in black.

**RC-1**

This study presents the first estimates of paired 10Be and 26Al cosmogenic nuclide denudation rates for catchments around Cuba, and these denudation rates are compared with chemical weathering fluxes derived from riverine solutes. The authors compare these two metrics across catchments within different lithologies (sedimentary, igneous, and metamorphic) and find that chemical weathering fluxes are often higher than total denudation fluxes. The authors interpret these results as evidence for deep chemical weathering that occurs below the upper couple of meters that where cosmogenic nuclides are produced. The high chemical weathering rates in this landscape are also consistent with other tropical landscapes around the world that generally find high chemical weathering rates, which indicates low rates of physical erosion. However, the long-term rates appear to generally be lower than short-term sediment yield fluxes, which the authors attribute to a period of enhanced agriculture during Soviet occupation.

We do not consider the rates we determine cosmogenically as denudation rates, which, in the literature, is typically defined as considering both physical and chemical mass loss. Because we do not know the depth at which chemical mass loss is occurring, the 10Be-determined rates likely miss some and perhaps much of the mass loss in solution for some samples. We will retool the early part of the paper to make this distinction clearer to readers.

The manuscript is well written and easy to follow, which is much appreciated! The goals of the manuscript are clear, and the background descriptions of methods such a cosmogenic nuclide dating were also well explained. The figures are all necessary and of good quality.

The results are certainly interesting and suggest that long-term denudation rates underestimate short-term denudation rates and chemical weathering fluxes.

We did not clearly articulate our findings and methods assumptions and have confused the reviewer. We do not consider the cosmogenic data as denudation rates, rather a lower limit on mass loss from the catchments. They are limits because chemical weathering and mass loss below the cosmogenic production zone (about 2 meters) is not detected by the cosmogenic data. In revision, we will work hard to clarify our interpretation of the data and the intrinsic limits and assumptions of the methods we use.

Most of my comments are minor, although I have two major comments related to the interpretation of the data and the decision to add the denudation and weathering fluxes, which I detail below. I hesitated between putting minor or major revisions for this manuscript, since I think they are actually moderate, but may also hinge upon further clarification of the methods and regional geology.

We will focus our revision on clarifying (1) our estimates of mass loss rates and (2) that measures of all three—sediment load, cosmogenic, and chemical load—have intrinsic assumptions and biases. We will clarify and better define the mass-loss terms we use throughout the manuscript.

**Moderate/Major Comments**

The Methods section, specifically related to the calculation of the weathering fluxes, requires more detail. As currently written, it's unclear to me whether the authors partitioned the concentrations of Ca and Na for silicate lithologies. If not, then this could very well call into question the interpretation of the chemical dissolution data as reflecting deep weathering that is not captured by the cosmogenic nuclide data. It may be that a clarification of the Methods section and added detailed to the regional geology would address my concern.

We will add detail and clarity to the methods section. The partitioning of Ca and Na for different lithologies is an interesting tool we had not previously considered. We will incorporate it into our analysis and use it as a way to interpret our data with the caveat that it is challenging to ensure accuracy as we partition solutes between carbonates, silicates, and evaporites because of the generalized, rudimentary geologic mapping in Cuba and because many basins drain multiple rock types. We have used quartz yields to estimate non-silicate percentages of sediment, although we understand these will over-estimate non-silicate percentage. We have generated plots similar to those in Eralanger's recent paper about mixed lithology orogens and placed our Cuba data in that context.

In the case that the full Ca and Na concentrations are used to determine weathering fluxes, the authors would be essentially comparing a "quartz" or silicate denudation rate with a chemical weathering flux that includes ions derived from both silicate and carbonate rocks. There is little description of the lithologies present in the study area, although Bierman et al. (2020) state that there is likely carbonate in all sampled basins. It would be important for the authors to clarify in what form carbonate is present (e.g. as a cement, as discrete layers within sedimentary rocks, as individual units, etc).

We will clarify in the manuscript that carbonate is present as discrete layers of rock, as cement, and as precipitates in the riverine sediment. Because our permits limited our sampling to particular points on rivers and did not include going upstream into the drainage basins we sampled, we cannot be more specific about rock types than what is mapped. However, we will incorporate the partioning you recommend and are using that to interpret our cosmogenic data in the context of variability in lithologies.

The reason this is important is because the authors need to understand whether the silicate and carbonate lithologies are weathering together—a cosmogenic denudation rates encompassing all lithologies could in this case be appropriate—or whether they denude separately.

This is not possible to tell with the scale of the mapping available. Because some of our quartz yields are < 10%, we suspect that quartz is present as stringers and lenses. We don't see towering outcrops of quartz-rich rocks, in fact we see very few outcrops at all. This is a soil mantled landscape.

In the latter case, it would make sense that the weathering fluxes might be altogether higher, particularly in the marine sedimentary units, which might reflect a large carbonate weathering flux that is largely absent from igneous and metamorphic rocks (except perhaps ophiolites)? Even for landscapes (e.g. New Zealand Southern Alps) where carbonate is present only in hydrothermal veins, the calculated carbonate weathering flux is still higher than the silicate weathering flux (Jacobson and Blum, 2003). So, it could make sense that the dissolution rates are higher than the denudation rates, since they are in fact reflecting all lithologies (carbonates, silicates, and maybe evaporites) while the denudation rates reflect only a portion of this. Of course, perhaps it's also a combination of deep weathering and carbonate weathering that are driving these rates. I'm not familiar with other studies in the tropics that have used chemical weathering from riverine solutes, so maybe there are comparisons that can be made there.

Perhaps one of our most important finding is that we have been able to isolate quartz sufficent for analysis even from basins mapped as entirely carbonate, which demonstrates that the basins are not underlain by pure carbonate rocks and that they contain detrital quartz. This wide of quartz does not allow different parts of the landscape (carbonate vs silicate) to weather at different rates. We will attempt to set upper and lower bounds and show them as a band on graphics. We will try a calculation for any samples with high Cl or SO4 or both and attribute all the Ca, Mg and Na to carbonate and evaporite weathering. This approach, however, disregards the measured high Si in some of those samples.

Without partitioning the weathering into silicates and carbonates, I'm not sure how the authors can exclude this possibility.

Because of mixed lithology basins we cannot confidently partition our rate to different rock types. But if there is a disconnect in weathering, then the topography should be evolving over time with carbonate/evaporite terrains lowering more quickly than the silicate terrains. We don't see that. Indeed, some of the highest erosion rates are in the steepest basins on the southern coast. These are silicate rich metamorphic rocks with low chemical weathering rates.

Since you have estimates for ion concentrations for precipitation, you should at least test how much this would alter your own data. You can also correct for cyclic inputs using global stoichiometric ratios for global average seawater. If these corrections are indeed minor, that would be justification to use the uncorrected data.

We will add this information to the manuscript and discuss its implications. It does not matter, however, because ionic flux in precipitation is low compared to flux in rivers.

Your only mention of active tectonics in Cuba is on Line 283. The active tectonics, particularly faults and fractures, could be important structure that facilitate the circulation of deeper groundwater and weathering, so I think more information needs to be added to the "Study Area" section that gives an overview of the tectonic setting.

We will do this. The active tectonics are to the east of the sampled basins though and there is no indication of substantial uplift that we are aware of in the area we sampled. There is fracturing at the outcrop scale but so few outcrops, we could not quantify. Eastern Cuba is much more tectonically active than central Cuba. We will examine what little structural mapping exists.

You also mention evaporite deposits in the basins, and your sentence on line 303 suggests that they are not exposed at the surface. If they are indeed only present in the subsurface, this suggests that you may have deeper circulation of groundwater in the region. Are there perhaps any springs in Cuba, thermal or otherwise?

Our permits restricted our sampling to pre-determined locations, all on rivers, and sampling methods (sediment and water), so cannot determine if evaporites crop out, but given the high mean annual precipitation and warm temperatures, outcrops seem unlikely. There are thermal and mineral springs of various compositions. We will cite this literature.

It's also not clear to me why the authors combine the dissolution rates with the denudation rates. The denudation rates already include the dissolution flux, since it is the total mass loss, so this seems somewhat redundant to me and goes beyond what a maximum denudation estimate could realistically be.

We clearly did not articulate well our most salient point, that cosmogenic 10Be rates for Cuba likely do not include all of the dissolution flux for all samples. We will carefully edit and expand the introduction to make this assumption and its violation in Cuba clear. If we could prove that all chemical weathering happened in the uppermost meter or two of regolith, then adding chemical would be redundant but in many, if not most basins, we suspect that some (or even most) chemical weathering is occurring deeper in the landscape, including at the base of the saprolite, as is the case in the Panola Mountain and Luquillo experimental watersheds. We clearly needed to improve our discussion of what is and isn't included in the cosmogenic rates, which only reflect mass loss in the uppermost several meters and will do that in the introduction and again in the discussion.

We will focus a part of the introduction on "What does the CRN content of alluvium actually measure, assuming that the steady-state and uniform distribution of quartz assumptions are reasonably correct?" The dissolved flux includes ions delivered from precipitation and the (soil) atmosphere, which do not represent mineral dissolution, and ions derived from the near surface weathering of rocks. In a perfect system you could probably specify where that weathering occurs and decide whether it was contributing directly to surface lowering (bedrock to saprolite to soil and then top-down soil erosion) or to the creation of caves and open

fractures that eventually will contribute to lowering. The alluvial CRN assumptions work best for granitic and high grade metamorphic rocks in steep landscapes, especially with high rates of uplift, and less well for layered carbonate and perhaps not at all for pure bedded gypsum and salt.

On Lines 357-364, you also compare the difference between your summed denudation and dissolution with the original denudation flux, and refer to that as the CEF. Perhaps I'm misunderstanding something, but how is your factor of increase comparable to the CEF? Riebe and Granger (2013) state that you need measurements of an insoluble element (usually Zr) to calculate CEF, which you also mention in section 2.2 but no insoluble elements were measured in this study.

We do not have the data to calculate the CEF because we were not allowed to sample bedrock in Cuba. We will revise this section to remove any suggestion that our work is equivalent to the CEF.

**Minor Comments**

The authors define terms for cosmogenic nuclide denudation rates as "sediment generation rates" and chemical weathering as "rock dissolution rates". These terms are not used consistently throughout the paper. I found example where "erosion" was used for the cosmogenic nuclide data, or to refer to physical erosion. I also found examples where "chemical denudation" or "chemical erosion" was used instead of "rock dissolution" or where "denudation" was used instead of "sediment generation rate".

This terminology was confusing and inconsistently used and we will replace it with "mass flux" for transparency and to avoid confusion and reduce assumption.

 "Sediment generation rate" to me implies physical erosion, rather than total mass loss or surface lowering, which is what the cosmogenic nuclide data represent.  I would highly recommend instead adopting the terms "denudation" and "chemical weathering", in order to avoid confusion, and to use them consistently throughout the paper.

As per above, we do not believe that all the cosmogenic data in central Cuba reflect denudation (total mass loss) and so have chosen to not adopt this nomenclature.

I would mention already in the Introduction that you also compare long-term denudation estimates from cosmogenic nuclides with short-term estimates from sediment yield fluxes. This point was on my mind for a long time as I read the paper, until I reached the discussion where you do in fact do this.

We will do this.

In general, I found that the figures could be referenced more throughout the paper when referring to results that they illustrate. Examples include line 248, where you could reference Figure 4 and Figure 7, and line 325, where you could reference Figure 4.

We will do this.

Methods. More detail can be given as to the specific methods used to calculate weathering fluxes. You mention using the West et al. (2005) method, although this study defines a couple of different methods for calculating weathering. If your method is equivalent to his cation weathering flux, it would be good to mention this and include an equation to make clear which cations and anions went into your calculations.

We will do this.

Supplement. The lithologic classifications shown on Figure 1 are not consistent with the categories given in Table 2 of the supplement, and "ultramafic" is missing altogether. It would be useful to include an additional row in the supplementary table as umbrella terms (with names equivalent to the categories in Figure 1) that would cover the various columns from the supplementary table.

We will do this.

The dashes for ranges of numbers (e.g. Line 227) should be En dashes.

We will do this.

When using terms such as low slope to describe a basin or other feature, they should be hyphenated, so "low-slope basins" (e.g. Line 267), as you've done for "low-relief topography" on line 269.

We will do this.

Your conclusion is the first time you state that you made the first measurements of cosmogenic nuclides in Cuba! You should definitely mention this in the introduction, since this is a nice contribution.

We will do this.

**Figures**

Figure 1. The legend for the geology is also somewhat unclear. The terms uC and pE are never explained in the text or the caption; I only figured them out when looking at the Supplement. It's also not clear to me from this Figure or from the supplement why the Upper Cretaceous marine deposits are differentiated from the Post-Eocene Marine deposits? If you

insist on keeping them separate, it would be important to mention in the text what the differences in composition are.

We keep these units apart because one contains evaporites and the other does not. We will add this explanation to the manuscript.

Is the term "undivided" supposed to be "undifferentiated"? Finally, it would be helpful to the reader to know what "Other" stands for, at least whether they are sedimentary rocks, igneous, or metamorphic, since some of your basins in the center of the field area appear to have a large part of the catchment that drains these areas.

The original map says "undivided" with no further explanation. Thus, we have left it.

The map figures would benefit from having the river networks includes and combining a hillshade map with the elevation DEM (as in Bierman et 2020). Otherwise, it not clear where the sampling location is for each basin, since I cannot tell where they flow! I would also like to see the locations of the discharge stations plotted on this map. If they are far away from the sampling location, it could be that they do not accurately reflect the discharge passing through the sampling location for the solutes. You also state that only 3 of your catchments had both sediment yield and cosmogenic nuclide measurements, so it would be good to see a map illustrating the locations where the other sediment yield measurements were made within your field area. Otherwise, it's difficult to make statements to this effect that sediment yield measurements are higher or lower than cosmogenic nuclide measurements, since there is overlap between the two datasets in your figure 8.

We will do this.

In your figure caption you write the panels with uppercase letters A and B, but there are shown as lowercase in the figures themselves.

We will do this.

In Figure 3C, could you put one color boundary that separates a ratio between rock dissolution and sediment generation at 1? That way the reader can more easily see where one process has a greater magnitude than the other.

We will do this.

**Line Comments**

Line 66. The authors use the term "at depth" in the paper. I understand that this implies depths greater than the upper couple of meters where cosmogenic nuclides are produced, but it would still be helpful to the reader to put some quantitative bounds on this term.

We will do this.

Lines 71-76. This is a very long sentence that was a bit hard for me to follow. I think it would benefit from being split.

We will do this.

Line 73. Are these the same data that were used in Bierman et al. (2020)? If so, I would cite them here, since they are already published.

We will do this.

Line 87-90 I understand what you mean with this sentence, but found it to be a bit misleading when I first read it, since it specifically refers to only rock dissolution. My suggestion would be to say "…cosmogenic nuclide rates cannot provide insight into **denudation processes** -such as rock dissolution- **occurring below depths of…** "

We will do this.

Line 121. I think a title for section 2.2 is missing

We will do this.

Line 241. Do you mean "Supplement T1" here?

We will do this.

Line 243. Figure 7 is also a nice illustration of the fact that anything that lies left of the 1:1 line has higher chemical weathering rates relative to total denudation rates!

We will add this observation to the text.

Line 265. You state that rock dissolution rates are strongly negatively correlated with slope, but

We will do this.

Line 284. I was a bit confused by the beginning of this sentence. Does the strong negative correlation refer to the Ollier study as well? It might be clearer if you start the sentence with a mention of this study so that it's clear to the reader you are not referring to your own results.

We will fix this wording.

Line 316. "of" is written twice.

We will do this.

Line 406. What do you mean when you say that "…neither consider the solutional component of denudation". If you are comparing denudation estimates, then they do include chemical weathering as part of the total mass loss.

We will rewrite this for clarity.

Line 406-408. It would be useful to cite the Bierman et al. (2020) study here to support your statement, since they made maps illustrating the locations of human influences.

We will cite this.

Line 431. I think you mean "of" here?

We will do this.

**RC2**

This manuscript presents new denudation rates from Cuba, and compares denudation measured with cosmogenic nuclides to solute loads in rivers from a previous study to infer substantial deep weathering in this low-relief, tropical setting. Using a paired nuclide approach, the authors are also able to constrain vertical soil mixing and quartz enrichment in some catchments. Overall, I found this paper to be well-written and interesting, and the implications for understanding global weathering fluxes and deep weathering make it both important and timely.

My concerns lie mostly with the way the data and analysis are presented, rather than with the underlying approach. I do think the required revisions are substantial,  but relatively straightforward (more like "moderate revisions" - I'm in agreement with the other reviewer on this).

**Major comments:**

Rock dissolution rates are inferred from solute loads in modern rivers and discharge measurements, which are from a previous study (Bierman et al 2020). The methods really aren't described here, and they need to be explained in more detail. In the 2020 paper, they're described very briefly in a supplemental file. As I understand it, solute fluxes were measured once, from samples taken at moderate discharges. These fluxes were then used to calculate weathering fluxes using average river discharges. However, if average discharges used to calculate fluxes were different from river discharges at the time samples were taken, or if fluxes vary susbstantially from rainy to dry seasons (which is very likely, given the seasonality of precipitation), these measurements could be way off. Incorporating some additional info on the variability in surface hydrology would be useful, and that variability should also be incorporated into uncertainties on the rock dissolution rates. I am concerned that there are no uncertainties

plotted on rock dissolution rates in the figures, which suggests that these fluxes are known much more precisely than is probably true.

We will add additional methodological information to the manuscript but hesitate to completely restate the methods and data in Bierman et al (2020) where the rates and methods are presented and all of the original data and calculations are presented in the supplement, which is freely available on line. We will add a discussion of the uncertainty in single sample collection and interpretation along with the caveat that when working in Cuba there was no other choice and our rationale for why having some measure of chemical load is better than having no measure of chemical load. We are not comfortable calculating formal uncertainties but have includes a discussion of the biases that could be reflected in doing such calculations. We do note that in many flow records, solute concentrations are either not dependent of flow or weakly dependent. We will discuss that in the text with citations. We also don't have actual discharge. We used modeled discharge, but recognize that has limitations as well. In some cases, there is a strong relationship between Q and C, but this is not generally the case, particularly for larger rivers.

It's also not clear how carbonate dissolution was handled, or how substantial evaporite contributions might be. These methodological details need to be included, as they will have substantial impact on how the high weathering fluxes are interpreted.

We will add this to the discussion and note that both of these represent mass flux out of the basin. We will separate samples with high Na and Cl and Ca and SO4 to assess the origin of sodium. Some of the ions could come from ppt and evaporation, but by balancing the anions (HCO3, SO4 and Cl) with Ca and Mg we will have a basis for assessing the relative contribution of evaporites.

I realize the authors are trying to use clear terminology, but the use of less-jargony terms here actually creates some confusion. The big one is "sediment generation", which is used for denudation measured with cosmogenic nuclides; this includes mass lost via rock dissolution in the top several meters of weathering profiles, so terming it sediment generation is potentially confusing because it could be interpreted as just the physical part of the flux. It is clearly defined early in the manuscript, but becomes problematic later in the discussion.

We will rework and revised our use of terminology in an attempt to more clearly communicate our approach and the underlying assumptions. We will focus on mass flux rather than sediment generation and we will avoid the use of the terms denudation and erosion.

There's a timescale mismatch between the solute fluxes and denudation rates, which isn't really discussed. Solute fluxes in rivers may represent a very brief snapshot, or may integrate over longer timescales depending on the groundwater residence time. Cosmogenic nuclides integrate over thousands of years in slow-eroding places, which is presumabley a much longer timescale.

We will add a discussion of time scales in the discussion section of the ms.

There is also potential for a spatial mismatch that makes these rates inappropriate to directly compare – the denudation rates reflect parts of the landscape that contribute quartz to rivers, and the solute fluxes reflect parts of the landscape that are dissolving. Given the diversity of rock types, including carbonates and mafic rocks with little quartz and high dissolution rates, the denudation and dissolution fluxes may be biased toward different parts of the landscape (and potentially different spatial extent). I don't think these potential mismatches are a manuscript-sinking problem, but it would be good to acknowledge the limitations and assumptions somewhere in the discussion.

We agree (as does Rev 1) and we will address the spatial issues in the ms with an additional paragraph in the discussion. The mapping in Cuba is not sufficient to determine how homogeneous the basins are lithologically but the relationship between solute loads and cosmogenic data suggests that rock type and quartz residence time are related and the lack of lithologic control on the landscape suggests (see response to rev 1) that quartz-bearing units or beds are interspersed with rocks that generate the solute load. We simply cannot know where the solute load is coming from beyond the partitioning calculations that Erlanger et al. used and that we also apply.

Groundwater – is it possible any of it exports directly to the sea, without going through rivers? In this case there may be even more mass loss via weathering.

Yes, it is possible but almost all of Cuba has extensive lowlands around the coast and so water table slopes are very low and we would thus expect fluxes to be moderate at most. We will add a sentence in the discussion.

The explanation of 26Al/10Be ratios via soil mixing is very interesting. Why are these particular soils mixed, and not others in the study? (Is there a mechanistic reason, or something about their position on the landscape? This is touched on briefly, and could be expanded if there's room – this is certainly not a requirement for publication, I just think it's interesting!)

Agree! We suspect it has to do with these particularly basins being both low slope and having extensive evaporite (or soluble?) deposits. We will add a paragraph discussing this finding.

I don't think it's appropriate to sum sediment generation and rock dissolution rates at the end – this is stated to be a max estimate, but discussed as though rates could actually be that high. This is why I'm not enthusiastic about the use of the term "sediment generation" – it implies that it's just the physical part of the flux, but it includes all the weathering fluxes that occur within the top couple of meters of weathering profiles (including all soil weathering). Summing sediment generation and rock dissolution counts near-surface weathering twice, which means these max denudation rates are an overestimate, and perhaps a substantial one.

We agree and will revise our approach to this in revision. See discussion in response to rev 1.

At line 405, the authors state that sediment yields and cosmogenic nuclide-derived denudation rates is directly comparable because the latter does not include mass loss due to dissolution. This is not true, as argued above. It is also interesting that the discrepancy between modern and long-term rates could reflect either changes in weathering depth or agricultural inputs (likely the latter), and teasing apart the two influences is potentially complicated in this deeply weathering setting.

Agree and will revise our discussion in light of this comment.

The discussion around Figure 7 could be expanded to include differences amongst rock types, which seem to be driving most of the variability. There's a lot more information in this plot than simply saying erosion and weathering aren't correlated in Cuba, which is how is currently reads.

Agree and will revise our discussion in light of this comment. XRD data (quantitative mineralogy) presented in an in prep paper support this idea that there are major differences in lithology and sediment composition. We have also referenced ideas in our previous paper (Bierman et al.) and have examined spatial clustering of the 10Be and chemical export results.

**Technical comments:**

This manuscript is very well-written and easy to follow, and the figures are generally quite good.

Line 121: 2.2 needs a title

We will do this.

Line 150: "Cuba's climate is tropical wet and dry" – it seems odd to describe something as both wet and dry like this. Maybe better described as strongly seasonal?

This is the formal Koppen climate classification. We will add explanation of seasonality

Figure 2: using a different color bar (or even trimming/stretching the grayscale) would make the elevation map more useful. Alternatively, a DEM hillshade overlain with the geology could simplify this to one panel, and make it easier to determine how lithology and topography correlate. "uC marine" and "pE marine" labels should be explained in the caption.

We will do this.

Fig. 6: trends in precip plots would be easier to see if the x-axis started at some higher value (750mm? 1000mm?) rather than zero. I appreciate that face that you didn't include trendlines on these plots, but just gave us the stats instead, especially where the trends are statistically significant but not neccessarily meaningful.

We will do this.

---

## Author Response (AR1)

**COLLEGE & CONSERVATORY**

**Geology Department**

Hella Wittmann, Editor
Geochronology

25 April 2022

To the Editor:

Please consider our revised manuscript, *Cosmogenic nuclide and solute flux data from central Cuban rivers emphasize the importance of both physical and chemical mass loss from tropical landscapes,* for publication in *Geochronology.*

We have extensively revised this manuscript following our submitted response to reviewers back in January. Substantial revisions include recalculating rock dissolution rates, including partitioning rates to infer dissolution rates from different lithologies, standardizing terminology, and rethinking the discussion section in light of the partitioned rock dissolution rates. In light of the extension revisions, we would like to request that reviewer 1 (Erlanger) be asked to review the paper again as many changes draw heavily on her work.

The manuscript is 7234 words long with seven figures, two tables, and 89 references. It also includes supporting material of two figures and thirteen tables. The data are under review at Pangaea (https://doi.pangaea.de/10.1594/PANGAEA.940043).

All authors have edited and approved the manuscript for submission. Please contact me if I can be of assistance in the review process.

Thank you for considering our work in *Geochronology.*

Sincerely,

Amanda Schmidt
Associate Professor of Geosciences

Our responses to the reviews are interspersed below. Our comments are in blue while the original text of the reviews is in black.

**RC-1**

This study presents the first estimates of paired 10Be and 26Al cosmogenic nuclide denudation rates for catchments around Cuba, and these denudation rates are compared with chemical weathering fluxes derived from riverine solutes. The authors compare these two metrics across catchments within different lithologies (sedimentary, igneous, and metamorphic) and find that chemical weathering fluxes are often higher than total denudation fluxes. The authors interpret these results as evidence for deep chemical weathering that occurs below the upper couple of meters that where cosmogenic nuclides are produced. The high chemical weathering rates in this landscape are also consistent with other tropical landscapes around the world that generally find high chemical weathering rates, which indicates low rates of physical erosion. However, the long-term rates appear to generally be lower than short-term sediment yield fluxes, which the authors attribute to a period of enhanced agriculture during Soviet occupation.

We do not consider the rates we determine cosmogenically as denudation rates, which, in the literature, are typically defined as including both physical and chemical mass loss. Because we do not know the depth at which chemical mass loss is occurring, the 10Be-determined rates likely miss some and perhaps much of the mass loss in solution for some samples. We have retooled the early part of the paper to make this distinction clearer to readers.

The manuscript is well written and easy to follow, which is much appreciated! The goals of the manuscript are clear, and the background descriptions of methods such a cosmogenic nuclide dating were also well explained. The figures are all necessary and of good quality.

The results are certainly interesting and suggest that long-term denudation rates underestimate short-term denudation rates and chemical weathering fluxes.

We did not clearly articulate our findings and methods assumptions and have confused the reviewer. We do not consider the cosmogenic data as denudation rates, rather a lower limit on mass loss from the catchments. They are limits because chemical weathering and mass loss below the cosmogenic production zone (about 2 meters) is not detected by the cosmogenic data.  In revision, we have worked hard to clarify our interpretation of the data and the intrinsic limits and assumptions of the methods we use.

Most of my comments are minor, although I have two major comments related to the interpretation of the data and the decision to add the denudation and weathering fluxes, which I detail below. I hesitated between putting minor or major revisions for this manuscript, since I think they are actually moderate, but may also hinge upon further clarification of the methods and regional geology.

We have focused our revision on clarifying (1) our estimates of mass loss rates and (2) that measures of all three—sediment load, cosmogenic, and chemical load—have intrinsic assumptions and biases. We have clarified and better defined the mass-loss terms we use throughout the manuscript.

Beyond this, we have completely recalculated the rock dissolution rates. As part of those calculations, we partitioned the rates into evaporite, carbonate, and silicate loss rates to have a better sense of what types of materials are dissolving and how that does (or doesn't) relate to cosmogenically derived erosion rates.

**Moderate/Major Comments**

The Methods section, specifically related to the calculation of the weathering fluxes, requires more detail. As currently written, it's unclear to me whether the authors partitioned the concentrations of Ca and Na for silicate lithologies. If not, then this could very well call into question the interpretation of the chemical dissolution data as reflecting deep weathering that is not captured by the cosmogenic nuclide data. It may be that a clarification of the Methods section and added detailed to the regional geology would address my concern.

We have added detail and clarity to the methods section. The partitioning of Ca and Na for different lithologies is an interesting tool we had not previously considered. We built on this tool and partitioned our dissolved load data into evaporite, carbonate, and silicate dissolution rates. However, it is hard to say for sure where the dissolved load comes from spatially because of the generalized, rudimentary geologic mapping in Cuba and because many basins drain multiple rock types. We have used quartz yields to estimate non-silicate percentages of sediment, although we understand these will over-estimate non-silicate percentage.

In the case that the full Ca and Na concentrations are used to determine weathering fluxes, the authors would be essentially comparing a "quartz" or silicate denudation rate with a chemical weathering flux that includes ions derived from both silicate and carbonate rocks. There is little description of the lithologies present in the study area, although Bierman et al. (2020) state that there is likely carbonate in all sampled basins. It would be important for the authors to clarify in what form carbonate is present (e.g. as a cement, as discrete layers within sedimentary rocks, as individual units, etc).

We note in the manuscript that carbonate is present as discrete layers of rock, as cement, and as precipitates in the riverine sediment. Because our permits limited our sampling to particular points on rivers and did not include going upstream into the drainage basins we sampled, we cannot be more specific about rock types than what is mapped. However, we have incorporated the partioning you recommend and are using that to interpret our cosmogenic data in the context of different lithologies.

The reason this is important is because the authors need to understand whether the silicate and carbonate lithologies are weathering together—a cosmogenic denudation rates

encompassing all lithologies could in this case be appropriate—or whether they denude separately.

This is not possible to tell with the scale of the mapping available. Because some of our quartz yields are < 10%, we suspect that quartz is present as stringers, lenses, and pods. These sorts of quartz distributions are common in low grade metamorphic rocks. We don't see towering outcrops of quartz-rich rocks, in fact we see very few outcrops at all. This is a soil mantled landscape.

In the latter case, it would make sense that the weathering fluxes might be altogether higher, particularly in the marine sedimentary units, which might reflect a large carbonate weathering flux that is largely absent from igneous and metamorphic rocks (except perhaps ophiolites)? Even for landscapes (e.g. New Zealand Southern Alps) where carbonate is present only in hydrothermal veins, the calculated carbonate weathering flux is still higher than the silicate weathering flux (Jacobson and Blum, 2003). So, it could make sense that the dissolution rates are higher than the denudation rates, since they are in fact reflecting all lithologies (carbonates, silicates, and maybe evaporites) while the denudation rates reflect only a portion of this. Of course, perhaps it's also a combination of deep weathering and carbonate weathering that are driving these rates. I'm not familiar with other studies in the tropics that have used chemical weathering from riverine solutes, so maybe there are comparisons that can be made there.

Perhaps one of our most important finding is that we have been able to isolate quartz sufficent for analysis even from basins mapped as entirely carbonate, which demonstrates that the basins are not underlain by pure carbonate rocks and that they contain detrital quartz. That being said, we did find that carbonate dissolution rates are the highest of the three categories for most basins. Without more detailed geologic mapping, we are unable to determine if the high carbonate dissolution rates has to do with fast rates of dissolution, large carbonate areas in most basins, or both. Likewise, the relatively lower evaporite dissolution rates in most basins could be because evaporites are dissolving slower than carbonates or because they are not as widespread. We are unable to distinguish between these two scenarios with the data we have.

Without partitioning the weathering into silicates and carbonates, I'm not sure how the authors can exclude this possibility.

With the partitioning of chemical weathering rates that we completed, as you suggested, we now argue that the landscape is maintaining its relief in part because of higher dissolution rates in flat areas and higher erosion rates in steeper areas. We have explored this in the revised discussion.

Since you have estimates for ion concentrations for precipitation, you should at least test how much this would alter your own data. You can also correct for cyclic inputs using global stoichiometric ratios for global average seawater. If these corrections are indeed minor, that would be justification to use the uncorrected data.

We have added this information to the manuscript and discuss its implications. Ionic flux in precipitation is low compared to flux in rivers.

Your only mention of active tectonics in Cuba is on Line 283. The active tectonics, particularly faults and fractures, could be important structure that facilitate the circulation of deeper groundwater and weathering, so I think more information needs to be added to the "Study Area" section that gives an overview of the tectonic setting.

We have now noted that the active tectonics in Cuba are to the east of the sampled basins and there is no indication of substantial uplift that we are aware of in the area we sampled. There is fracturing at the outcrop scale but there are too few outcrops to quantify. Eastern Cuba is much more tectonically active than central Cuba. We were unable to find any structural maps for Cuba. The publicly available mapping is limited to USGS oil exploration data.

You also mention evaporite deposits in the basins, and your sentence on line 303 suggests that they are not exposed at the surface. If they are indeed only present in the subsurface, this suggests that you may have deeper circulation of groundwater in the region. Are there perhaps any springs in Cuba, thermal or otherwise?

Our permits restricted our sampling to pre-determined locations, all on rivers, and sampling methods (sediment and water), so cannot determine if evaporites crop out, but given the high mean annual precipitation and warm temperatures, outcrops seem unlikely. There are thermal and mineral springs of various compositions and we have cited this literature.

It's also not clear to me why the authors combine the dissolution rates with the denudation rates. The denudation rates already include the dissolution flux, since it is the total mass loss, so this seems somewhat redundant to me and goes beyond what a maximum denudation estimate could realistically be.

We had not articulated well our most salient point, that cosmogenic 10Be rates for Cuba likely do not include all of the dissolution flux for all samples. We have carefully edited and expanded the introduction to make this assumption and its violation in Cuba clear. If we could prove that all chemical weathering happened in the uppermost meter or two of regolith, then adding the chemical flux would be redundant. In many, if not most basins, we suspect that some (or even most) chemical weathering is occurring deeper in the landscape, including at the base of the saprolite, as is the case in the Panola Mountain (Georgia) and Luquillo (Puerto Rico) experimental watersheds. We have clarified (in our Introduction and Discussion) what is and isn't included in the cosmogenic rates, which only reflect mass loss in the uppermost several meters.

We focus a part of the introduction on "What does the CRN content of alluvium actually measure, assuming that the steady-state and uniform distribution of quartz assumptions are reasonably correct?" The dissolved flux includes ions delivered from precipitation and the (soil) atmosphere, which do not represent mineral dissolution, and ions derived from the near

surface weathering of rocks. In a perfect system you could probably specify where that weathering occurs and decide whether it was contributing directly to surface lowering (bedrock to saprolite to soil and then top-down soil erosion) or to the creation of caves and open fractures that eventually will contribute to lowering. The alluvial CRN assumptions work best for granitic and high grade metamorphic rocks in steep landscapes, especially with high rates of uplift, and less well for layered carbonate and perhaps not at all for pure bedded gypsum and salt.

On Lines 357-364, you also compare the difference between your summed denudation and dissolution with the original denudation flux, and refer to that as the CEF. Perhaps I'm misunderstanding something, but how is your factor of increase comparable to the CEF? Riebe and Granger (2013) state that you need measurements of an insoluble element (usually Zr) to calculate CEF, which you also mention in section 2.2 but no insoluble elements were measured in this study.

We do not have the data to calculate the CEF because we our permits limited sampling to river sediment and water. We revised this section to remove any suggestion that our work is equivalent to the CEF.

**Minor Comments**

The authors define terms for cosmogenic nuclide denudation rates as "sediment generation rates" and chemical weathering as "rock dissolution rates". These terms are not used consistently throughout the paper. I found example where "erosion" was used for the cosmogenic nuclide data, or to refer to physical erosion. I also found examples where "chemical denudation" or "chemical erosion" was used instead of "rock dissolution" or where "denudation" was used instead of "sediment generation rate".

This terminology was confusing and inconsistently used and we have replaced it with "mass loss" for transparency and to avoid confusion and reduce assumption.

 "Sediment generation rate" to me implies physical erosion, rather than total mass loss or surface lowering, which is what the cosmogenic nuclide data represent.  I would highly recommend instead adopting the terms "denudation" and "chemical weathering", in order to avoid confusion, and to use them consistently throughout the paper.

As per above, we do not believe that all the cosmogenic data in central Cuba reflect denudation (total mass loss) and chose to not adopt this nomenclature.

I would mention already in the Introduction that you also compare long-term denudation estimates from cosmogenic nuclides with short-term estimates from sediment yield fluxes. This point was on my mind for a long time as I read the paper, until I reached the discussion where you do in fact do this.

We have done this.

In general, I found that the figures could be referenced more throughout the paper when referring to results that they illustrate. Examples include line 248, where you could reference Figure 4 and Figure 7, and line 325, where you could reference Figure 4.

We have done this.

Methods. More detail can be given as to the specific methods used to calculate weathering fluxes. You mention using the West et al. (2005) method, although this study defines a couple of different methods for calculating weathering. If your method is equivalent to his cation weathering flux, it would be good to mention this and include an equation to make clear which cations and anions went into your calculations.

We will have done this.

Supplement. The lithologic classifications shown on Figure 1 are not consistent with the categories given in Table 2 of the supplement, and "ultramafic" is missing altogether. It would be useful to include an additional row in the supplementary table as umbrella terms (with names equivalent to the categories in Figure 1) that would cover the various columns from the supplementary table.

We have done this.

The dashes for ranges of numbers (e.g. Line 227) should be En dashes.

We have done this.

When using terms such as low slope to describe a basin or other feature, they should be hyphenated, so "low-slope basins" (e.g.  Line 267), as you've done for "low-relief topography" on line 269.

We have done this.

Your conclusion is the first time you state that you made the first measurements of cosmogenic nuclides in Cuba! You should definitely mention this in the introduction, since this is a nice contribution.

We have done this.

**Figures**

Figure 1. The legend for the geology is also somewhat unclear. The terms uC and pE are never explained in the text or the caption; I only figured them out when looking at the

Supplement.  It's also not clear to me from this Figure or from the supplement why the Upper Cretaceous marine deposits are differentiated from the Post-Eocene Marine deposits? If you insist on keeping them separate, it would be important to mention in the text what the differences in composition are.

We distinguish these units because one contains evaporites and the other does not. We added this explanation to the manuscript.

Is the term "undivided" supposed to be "undifferentiated"? Finally, it would be helpful to the reader to know what "Other" stands for, at least whether they are sedimentary rocks, igneous, or metamorphic, since some of your basins in the center of the field area appear to have a large part of the catchment that drains these areas.

The original map says "undivided" with no further explanation. Thus, we have left it.

The map figures would benefit from having the river networks includes and combining a hillshade map with the elevation DEM (as in Bierman et 2020). Otherwise, it not clear where the sampling location is for each basin, since I cannot tell where they flow! I would also like to see the locations of the discharge stations plotted on this map. If they are far away from the sampling location, it could be that they do not accurately reflect the discharge passing through the sampling location for the solutes. You also state that only 3 of your catchments had both sediment yield and cosmogenic nuclide measurements, so it would be good to see a map illustrating the locations where the other sediment yield measurements were made within your field area. Otherwise, it's difficult to make statements to this effect that sediment yield measurements are higher or lower than cosmogenic nuclide measurements, since there is overlap between the two datasets in your figure 8.

We have done this.

In your figure caption you write the panels with uppercase letters A and B, but there are shown as lowercase in the figures themselves.

We have fixed this.

In Figure 3C, could you put one color boundary that separates a ratio between rock dissolution and sediment generation at 1? That way the reader can more easily see where one process has a greater magnitude than the other.

We have done this (the lightest pink is the ratio <1).

**Line Comments**

Line 66. The authors use the term "at depth" in the paper. I understand that this implies depths greater than the upper couple of meters where cosmogenic nuclides are produced, but it would still be helpful to the reader to put some quantitative bounds on this term.

We have done this.

Lines 71-76. This is a very long sentence that was a bit hard for me to follow.  I think it would benefit from being split.

We fixed this.

Line 73. Are these the same data that were used in Bierman et al. (2020)? If so, I would cite them here, since they are already published.

We have done this.

Line 87-90 I understand what you mean with this sentence, but found it to be a bit misleading when I first read it, since it specifically refers to only rock dissolution. My suggestion would be to say "…cosmogenic nuclide rates cannot provide insight into **denudation processes** -such as rock dissolution- **occurring below depths of…** "

We have done this.

Line 121. I think a title for section 2.2 is missing

We have done this.

Line 241. Do you mean "Supplement T1" here?

We have done this.

Line 243.  Figure 7 is also a nice illustration of the fact that anything that lies left of the 1:1 line has higher chemical weathering rates relative to total denudation rates!

We have added this observation to the text.

Line 265. You state that rock dissolution rates are strongly negatively correlated with slope, but

We have done this.

Line 284. I was a bit confused by the beginning of this sentence. Does the strong negative correlation refer to the Ollier study as well? It might be clearer if you start the sentence with a mention of this study so that it's clear to the reader you are not referring to your own results.

We have fixed this wording.

Line 316. "of" is written twice.

We have done this.

Line 406. What do you mean when you say that "…neither consider the solutional component of denudation". If you are comparing denudation estimates, then they do include chemical weathering as part of the total mass loss.

We have rewritten this for clarity.

Line 406-408. It would be useful to cite the Bierman et al. (2020) study here to support your statement, since they made maps illustrating the locations of human influences.

We have cited this.

Line 431. I think you mean "of" here?

We have done this.

**RC2**

This manuscript presents new denudation rates from Cuba, and compares denudation measured with cosmogenic nuclides to solute loads in rivers from a previous study to infer substantial deep weathering in this low-relief, tropical setting. Using a paired nuclide approach, the authors are also able to constrain vertical soil mixing and quartz enrichment in some catchments. Overall, I found this paper to be well-written and interesting, and the implications for understanding global weathering fluxes and deep weathering make it both important and timely.

My concerns lie mostly with the way the data and analysis are presented, rather than with the underlying approach. I do think the required revisions are substantial, but relatively straightforward (more like "moderate revisions" - I'm in agreement with the other reviewer on this).

**Major comments:**

Rock dissolution rates are inferred from solute loads in modern rivers and discharge measurements, which are from a previous study (Bierman et al 2020). The methods really aren't described here, and they need to be explained in more detail. In the 2020 paper, they're described very briefly in a supplemental file. As I understand it, solute fluxes were measured once, from samples taken at moderate discharges. These fluxes were then used to calculate

weathering fluxes using average river discharges. However, if average discharges used to calculate fluxes were different from river discharges at the time samples were taken, or if fluxes vary susbstantially from rainy to dry seasons (which is very likely, given the seasonality of precipitation), these measurements could be way off. Incorporating some additional info on the variability in surface hydrology would be useful, and that variability should also be incorporated into uncertainties on the rock dissolution rates. I am concerned that there are no uncertainties plotted on rock dissolution rates in the figures, which suggests that these fluxes are known much more precisely than is probably true.

We have added additional methodological information to the manuscript but hesitate to completely restate the methods and data in Bierman et al (2020) where the rates and methods are presented and all of the original data and calculations are presented in the supplement, which is freely available on line. We added a discussion of the uncertainty in single sample collection and interpretation along with the caveat that when working in Cuba there was no other choice and our rationale that having some measure of chemical load is better than having no measure of chemical load. We are not comfortable calculating formal uncertainties but have includes a discussion of the biases that could be reflected in doing such calculations. We do note that in many flow records, most solute concentrations are either not dependent on flow or weakly dependent. We discuss these points in the text, with citations. We also don't have actual discharge. We used modeled discharge, but recognize that this approach has limitations as well. In some cases, there is a strong relationship between Q and C, but this is not generally the case, particularly for larger rivers. In addition, we completely recalculated the rock dissolution rates, so that is now explained fully.

It's also not clear how carbonate dissolution was handled, or how substantial evaporite contributions might be. These methodological details need to be included, as they will have substantial impact on how the high weathering fluxes are interpreted.

We have added this information to the discussion and note that both represent mass flux out of the basin. We separated samples with high Na and Cl and Ca and SO4 to assess the origin of sodium and by balancing the anions (HCO3, SO4 and Cl) with Ca and Mg we were able to assess the relative contribution of silicate, carbonate and evaporite weathering.

I realize the authors are trying to use clear terminology, but the use of less-jargony terms here actually creates some confusion. The big one is "sediment generation", which is used for denudation measured with cosmogenic nuclides; this includes mass lost via rock dissolution in the top several meters of weathering profiles, so terming it sediment generation is potentially confusing because it could be interpreted as just the physical part of the flux. It is clearly defined early in the manuscript, but becomes problematic later in the discussion.

We have reworked and revised our use of terminology in an attempt to more clearly communicate our approach and the underlying assumptions. We use "mass loss" rather than "sediment generation" and we avoid using the terms denudation and erosion except as clearly defined in the introduction.

There's a timescale mismatch between the solute fluxes and denudation rates, which isn't really discussed. Solute fluxes in rivers may represent a very brief snapshot, or may integrate over longer timescales depending on the groundwater residence time. Cosmogenic nuclides integrate over thousands of years in slow-eroding places, which is presumabley a much longer timescale.

We have added a discussion of time scales in the discussion section of the text.

There is also potential for a spatial mismatch that makes these rates inappropriate to directly compare – the denudation rates reflect parts of the landscape that contribute quartz to rivers, and the solute fluxes reflect parts of the landscape that are dissolving. Given the diversity of rock types, including carbonates and mafic rocks with little quartz and high dissolution rates, the denudation and dissolution fluxes may be biased toward different parts of the landscape (and potentially different spatial extent). I don't think these potential mismatches are a manuscript-sinking problem, but it would be good to acknowledge the limitations and assumptions somewhere in the discussion.

We agree (as does Rev 1) and we have addressed the spatial issues in the text with an additional paragraph in the discussion. Geologic mapping in Cuba is not sufficient to determine how homogeneous the basins are lithologically but the relationship between solute loads and cosmogenic data suggests that rock type and quartz residence time are related and the lack of lithologic control on the landscape suggests (see response to rev 1) that quartz-bearing units or beds are interspersed with rocks that generate the solute load. We simply cannot know where the solute load is coming from beyond the partitioning calculations that Erlanger et al. used and that we have applied.

Groundwater – is it possible any of it exports directly to the sea, without going through rivers? In this case there may be even more mass loss via weathering.

Yes, it is possible but almost all of Cuba has extensive lowlands around the coast and so water table slopes are very low and we thus expect fluxes to be moderate at most. We added a sentence in the discussion.

The explanation of 26Al/10Be ratios via soil mixing is very interesting. Why are these particular soils mixed, and not others in the study? (Is there a mechanistic reason, or something about their position on the landscape? This is touched on briefly, and could be expanded if there's room – this is certainly not a requirement for publication, I just think it's interesting!)

Agree! We suspect it has to do with these particular basins being both low slope and having extensive evaporite (or soluble?) deposits. We added a paragraph discussing this finding.

I don't think it's appropriate to sum sediment generation and rock dissolution rates at the end – this is stated to be a max estimate, but discussed as though rates could actually be that high. This is why I'm not enthusiastic about the use of the term "sediment generation" – it implies

that it's just the physical part of the flux, but it includes all the weathering fluxes that occur within the top couple of meters of weathering profiles (including all soil weathering). Summing sediment generation and rock dissolution counts near-surface weathering twice, which means these max denudation rates are an overestimate, and perhaps a substantial one.

We agree and revised our approach to this in revision. See discussion in response to rev 1.

At line 405, the authors state that sediment yields and cosmogenic nuclide-derived denudation rates is directly comparable because the latter does not include mass loss due to dissolution. This is not true, as argued above. It is also interesting that the discrepancy between modern and long-term rates could reflect either changes in weathering depth or agricultural inputs (likely the latter), and teasing apart the two influences is potentially complicated in this deeply weathering setting.

Agree and revised our discussion in light of this comment.

The discussion around Figure 7 could be expanded to include differences amongst rock types, which seem to be driving most of the variability. There's a lot more information in this plot than simply saying erosion and weathering aren't correlated in Cuba, which is how is currently reads.

Agree and revised our discussion in light of this comment. XRD data (quantitative mineralogy) presented in an in prep paper support the idea that there are major differences in lithology and sediment composition. We have also referenced ideas in our previous paper (Bierman et al.) and have examined spatial clustering of the 10Be and chemical export results.

**Technical comments:**

This manuscript is very well-written and easy to follow, and the figures are generally quite good.

Line 121: 2.2 needs a title

We have done this.

Line 150: "Cuba's climate is tropical wet and dry" – it seems odd to describe something as both wet and dry like this. Maybe better described as strongly seasonal?

This is the formal Koppen climate classification. We added an explanation of seasonality

Figure 2: using a different color bar (or even trimming/stretching the grayscale) would make the elevation map more useful. Alternatively, a DEM hillshade overlain with the geology could simplify this to one panel, and make it easier to determine how lithology and topography correlate. "uC marine" and "pE marine" labels should be explained in the caption.

We have done this.

Fig. 6: trends in precip plots would be easier to see if the x-axis started at some higher value (750mm? 1000mm?) rather than zero. I appreciate that face that you didn't include trendlines on these plots, but just gave us the stats instead, especially where the trends are statistically significant but not neccessarily meaningful.

We have done this.

---

## Editor Decision (ED1)

Dear Dr. Schmidt et al.,

I have now received two reviews and your correspondence to your paper "Cosmogenic nuclide and solute flux data from central Cuba emphasize the importance of both physical and chemical denudation in highly weathered landscapes".

Reviewer #1 suggests that revisions are minor with tendencies to moderate revisions, while reviewer #2 suggest major revisions. My own tendency goes into the same direction as Reviewer #1, such that I will decide after seeing the revised (track-changed) manuscript if another round of reviews will be necessary.

Common issues that both reviewers point out relate to confusing terminology, and calculation of weathering rates from dissolved load measurements. I think the authors received here good suggestions on how to improve the paper, and the associated changes are probably more towards moderate than actually "major" in terms of scientific changes.

I also have issues with summing up "sediment generation rates" and rock dissolution rates, as these two approaches do cover very different temporal scales. I am also a bit intrigued by the fact that you seem to observe highest rock dissolution rates but lowest cosmo rates for sedimentary rocks. I would have expected high dissolution rates for the volcanic and carbonate-bearing rocks instead, and would suggest that you discuss reasons for this in more detail, one of them potentially including a bias from using quartz-based nuclides in rocks where quartz is of minor abundance. I understand that taking these lithologies apart is very difficult, because of restricted field access, but perhaps the approach suggested by Reviewer #1 on Ca and Na partitioning might help here. However, you mention several times in the MS that lithologic control is very important in this landscape- hence, giving more information about the different rock types is essential for me.

All the best, Hella Wittmann-Oelze

---

## Author Response (AR2)

**Geology Department**

Hella Wittmann, Editor
Geochronology

20 May 2022

To Dr. Wittmann:

Please consider our revised manuscript, *Cosmogenic nuclide and solute flux data from central Cuban rivers emphasize the importance of both physical and chemical mass loss from tropical landscapes*, for publication in *Geochronology*.

We have completed the few minor revisions that you requested and attach a response to reviews to this letter, as well as a tracked changes version of the manuscript.

The manuscript is 7250 words long with seven figures, two tables, and 89 references. It also includes supporting material of two figures and thirteen tables. The data have been reviewed at Pangaea (https://doi.org/10.1594/PANGAEA.940051).

All authors have edited and approved the manuscript for submission. Please contact me if I can be of assistance in the review process.

Thank you for considering our work in *Geochronology*.

Sincerely,

Amanda Schmidt
Associate Professor of Geosciences

Dear Dr. Schmidt et al.,

as requested, I had sent out the paper to Reviewer Erica Erlanger, but she declined the invitation. However, Claire Lukens had another look at the paper and she was highly satisfied with the changes.
As for myself, I have the following suggestions to you (below). I will accept the paper now, but please don´t forget to incorporate these suggestions during the proof stage. Thank you. Overall, I agree with Dr. Lukens that the paper is now much improved over the previous version and I really appreciate your efforts!

With best regards, Hella

Thank you for your comments and for your patience with us. Our responses to your feedback are below in blue.

Line 29 Awkward wording. Perhaps rephrase to "in this environment, landscape-scale mass loss…"  Fixed.

Line 158 Typo "Blanckenburg" (here and elsewhere)  Fixed.

Line 280: Please indicate in the Table that for samples that have low Al/Be ratios, the rates of erosion are overestimates and should not be taken for true values. Some of the column headers are not conclusive.  Added footnote to the table.

Line 294 Please insert "weakly" before "positively correlated".  Fixed.

Line 323: include the year after Linari et al.  Fixed.

Line 350: now here is the place to state that in non-tropical catchments, chemical weathering rates from dissolved loads are often much lower than the rates obtained from cosmogenic nuclides, and briefly discuss integration time scale differences  See below.

Line 385 "longer"- relative to what? E.g. provide time frame.  Fixed.

Line 450ff.: I am a bit unsatisfied with the fact that the integration time scale difference between these methods is nowhere discussed before summing them up. Please include a short paragraph.

The comments on line 350 and 450 are related, so we address them here:
We are uncomfortable adding the sentence about non-tropical comparisons between rock dissolution rates and 10Be-derived erosion rates. There are scenarios in climate regions all over the world where we can imagine mismatches of these rates (for example, the Canadian shield is totally flat, so 10Be-derived erosion rates will be low but there must be rock dissolution rates). We have added a more generic statement about types of environments that favor rapid rock dissolution rates.

We also added a paragraph before the paragraph in question explaining the timescale differences.

We revisit that idea in a new paragraph added around the line 450 paragraph you mention.

**Report #1**

Submitted on 17 May 2022
Referee #2: Claire E Lukens, clukens@ucmerced.edu

**Anonymous during peer-review:** Yes **No**
**Anonymous in acknowledgements of published article:** Yes **No**

**Recommendation to the editor**

**1) Scientific significance**
Does the manuscript represent a substantial contribution to scientific progress within the scope of this journal (substantial new concepts, theories, methods, or data)?

**Excellent** Good Fair Poor

**2) Scientific quality**
Are the scientific approach and applied methods valid? Are the results discussed in an appropriate and balanced way (consideration of related work, including appropriate references)?

**Excellent** Good Fair Poor

**3) Presentation quality**
Are the scientific results and conclusions presented in a clear, concise, and well structured way (number and quality of figures/tables, appropriate use of English language)?

**Excellent** Good Fair Poor

For final publication, the manuscript should be
**accepted as is.**
accepted subject to **technical corrections**.
accepted subject to **minor revisions**.
reconsidered after **major revisions**:
**rejected**.

**Were a revised manuscript to be sent for another round of reviews:**
I would be willing to review the revised manuscript.
**I would not be willing to review the revised manuscript.**

**Suggestions for revision or reasons for rejection (will be published if the paper is accepted for final publication)**
Thanks to the authors for the incredibly thorough and thoughtful revision. I am satisfied that my initial comments were addressed, and I have no further suggestions for improvement. I look forward to seeing this in print!

Thank you for taking the time to read our manuscript again and for the helpful comments on the first version of the paper.

[revised manuscript text omitted]

---

## Editor Decision (ED2)

Dear Dr. Schmidt et al.,

as requested, I had sent out the paper to Reviewer Erica Erlanger, but she declined the invitation. However, Claire Lukens had another look at the paper and she was highly satisfied with the changes.
As for myself, I have the following suggestions to you (below). I will accept the paper now, but please don´t forget to incorporate these suggestions during the proof stage. Thank you. Overall, I agree with Dr. Lukens that the paper is now much improved over the previous version and I really appreciate your efforts!

With best regards, Hella

Line 29 Awkward wording. Perhaps rephrase to "in this environment, landscape-scale mass loss…"
Line 158 Typo "Blanckenburg" (here and elsewhere)
Line 280: Please indicate in the Table that for samples that have low Al/Be ratios, the rates of erosion are overestimates and should not be taken for true values. Some of the column headers are not conclusive.
Line 294 Please insert "weakly" before "positively correlated".
Line 323: include the year after Linari et al.
Line 350: now here is the place to state that in non-tropical catchments, chemical weathering rates from dissolved loads are often much lower than the rates obtained from cosmogenic nuclides, and briefly discuss integration time scale differences
Line 385 "longer"- relative to what? E.g. provide time frame.
Line 450ff.: I am a bit unsatisfied with the fact that the integration time scale difference between these methods is nowhere discussed before summing them up. Please include a short paragraph.